# Combined numerical and experimental study of microstructure and permeability in porous granular media

Philipp Eichheimer[1], Marcel Thielmann[1], Wakana Fujita[2], Gregor J. Golabek[1], Michihiko Nakamura[2], Satoshi Okumura[2], Takayuki Nakatani[2], and Maximilian O. Kottwitz[3]

[1]Bayerisches Geoinstitut, University of Bayreuth, Universitätsstrasse 30, 95447 Bayreuth, Germany
[2]Department of Earth Science, Tohoku University, 6-3, Aramaki Aza-Aoba, Aoba-ku, Sendai 980-8578, Japan
[3]Institute of Geoscience, Johannes Gutenberg University, Johann-Joachim-Becher-Weg 21, 55128 Mainz, Germany

**Correspondence:** Philipp Eichheimer (Philipp.Eichheimer@uni-bayreuth.de)

**Abstract.** Fluid flow on different scales is of interest for several Earth science disciplines like petrophysics, hydrogeology and volcanology. To parameterize fluid flow in large-scale numerical simulations (e.g. groundwater and volcanic systems), flow properties on the microscale need to be considered. For this purpose experimental and numerical investigations of flow through porous media over a wide range of porosities are necessary. In the present study we sinter glass bead media with various porosities and measure the permeability experimentally. The microstructure, namely effective porosity and effective specific surface, is investigated using image processing. We determine flow properties like tortuosity and permeability using numerical simulations. We test different parameterizations for isotropic low porosity media on their potential to predict permeability by comparing their estimations to computed and experimentally measured values.

## 1 Introduction

The understanding of transport and storage of geological fluids in sediments, crust and mantle is of major importance for several Earth science disciplines including volcanology, hydrology and petroleum geoscience (e.g. Manwart et al., 2002; Ramandi et al., 2017; Honarpour, 2018). In volcanic settings melt segregation from partially molten rocks controls the magma chemistry, and outgassing of magmas influences both magma ascent and eruption explosivity (Collinson and Neuberg, 2012; Lamur et al., 2017; Mueller et al., 2005). In hydrogeology fluid flow affects ground water exploitation and protection (Domenico and Schwartz, 1998; Hölting and Coldewey, 2019), whereas in petroleum geoscience it controls oil recovery efficiency (Suleimanov et al., 2011; Hendraningrat et al., 2013; Zhang et al., 2014).

A key parameter for fluid flow is permeability. Permeability estimations have been performed on several scales ranging from pore scale (Brace, 1980) to macroscale (Fehn and Cathles, 1979; Norton and Taylor Jr, 1979; Gleeson and Ingebritsen, 2016). As the permeability on the macroscale is a function of its microstructure it is necessary to accurately predict permeability based on microscale properties (Mostaghimi et al., 2013). To achieve this goal, various experimental and numerical approaches have been developed over the years (e.g. Keehm, 2003; Andrä et al., 2013a; Gerke et al., 2018; Saxena et al., 2017).

Assuming laminar flow (Bear, 1988; Matyka et al., 2008), flow through porous media can be described using Darcy's law (Darcy, 1856), which relates the fluid flux $Q$ to an applied pressure difference $\Delta P$

$$Q = -\frac{kA\Delta P}{\eta L}, \tag{1}$$

where $k$ is the permeability, $A$ is the cross sectional area, $\eta$ is the fluid viscosity and $L$ is the length of the domain.

Accurately determining and predicting permeability is thus of crucial importance to quantify fluid fluxes in porous media. Until today it remains challenging to relate permeability to the microstructure of porous media. This has resulted in numerous parameterizations developed for different materials and structures (Kozeny, 1927; Carman, 1937, 1956; Martys et al., 1994; Revil and Cathles III, 1999; Garcia et al., 2009).

A first simple capillary model to predict the permeability of a porous medium was proposed by Kozeny (1927)

$$k = k_0 \frac{\phi^3}{S^2}, \tag{2}$$

where $k_0$ is the dimensionless Kozeny constant depending on the channel geometry (e.g. $k_0 = 0.5$ for cylindrical capillaries), $\phi$ is the porosity and $S$ is the specific surface area (ratio of exposed surface area to bulk volume). Later this relation was extended by Carman (1937, 1956), to predict fluid flow through a granular bed with a given microstructure. To account for the effect of the microstructure on fluid flow, Carman (1937, 1956) introduced the term tortuosity, which he defined as the ratio of effective flow path $L_e$ to a straight path $L$.

$$\tau = \frac{L_e}{L} \tag{3}$$

Introducing this relation into eq.(2) leads to the well-known Kozeny-Carman equation:

$$k = k_0 \frac{\phi^3}{\tau^2 S^2}, \tag{4}$$

Using experimental data, Carman (1956) determined that tortuosity $\tau$ is $\approx \sqrt{2}$. Today, the Kozeny-Carman equation - or variants thereof - is widely used in volcanology (Klug and Cashman, 1996; Mueller et al., 2005; Miller et al., 2014), hydrogeology (Wang et al., 2017; Taheri et al., 2017), two-/multi-phase flow studies (Wu et al., 2012; Keller and Katz, 2016; Keller and Suckale, 2019) and soil sciences (Chapuis and Aubertin, 2003; Ren et al., 2016). The Kozeny-Carman equation was derived assuming that the medium consists only of continuous curved channels with constant cross-section (Carman, 1937; Bear, 1988). However, in porous media pathways most likely do not obey these assumptions. Applying this equation to porous media therefore remains challenging and in some cases fails for low porosities (Bernabe et al., 1982; Bourbie et al., 1992) or mixtures of different shapes and material sizes (Carman, 1937; Wyllie and Gregory, 1955). Consequently, alternative permeability parameterizations have been developed by different authors (Martys et al., 1994; Revil and Cathles III, 1999; Garcia et al., 2009).

Using numerical modeling, Martys et al. (1994) derived a universal scaling law for various overlapping and non-overlapping sphere packings which reads as:

$$k = \frac{2(1 - \phi - \phi_c)}{S^2}(\phi - \phi_c)^f, \tag{5}$$

with $f = 4.2$ and $\phi_c$ being the critical porosity, below which no connected pore space exists. They showed that eq.(5) is valid for a variety of porous media including mono-sized sphere packings, glass bead samples and experimentally measured sandstones. Despite the predictive power of this parameterization it might not give reasonable estimations for permeability in case the porous medium consists of rough surfaces and large isolated regions (voids).

The study of Revil and Cathles III (1999) used electrical parameters to derive the permeability of different types of shaly sands, i.e., the permeability of a clay-free sand and the permeability of a pure shale. By using electrical parameters which separate pore throat from total porosity and effective from total hydraulic radius, Revil and Cathles III (1999) were able to improve the Kozeny-Carman relation, being only dependent on grain size. In a first step the authors developed a model for the permeability of a clay-free sand as a function of the grain diameter, the porosity, and the electrical cementation exponent reading as:

$$\Lambda = \frac{R^2}{2m^2 F^3}, \tag{6}$$

with $\Lambda$ being the effective electrical pore radius, $R$ being the grain radius, $m$ being the cementation exponent and $F$ being the formation factor. Using the relation of the formation factor to porosity by Archie's law $F = \phi^{-m}$ (Waxman and Smits, 1968), $m = 1.8$ (Waxman and Smits, 1968) and $d = 2R$ for the grain diameter the authors derived a permeability parameterization for natural sandstones:

$$k = \frac{d^2 \phi^{5.1}}{24}, \tag{7}$$

which is in good agreement with experimentally measured data by Berg (1975).

Based on numerical simulations of fluid flow in polydisperse grain packings with irregular shapes, Garcia et al. (2009) proposed an alternative parameterization by fitting the numerical results with the following equation:

$$k = \phi^{0.11} D^2, \tag{8}$$

where $D^2$ is the squared harmonic mean diameter of the grains. They also showed that this parameterization also fits experimental results quite well and concluded that grain shape and size polydispersity have a small but noticeable effect on permeability.

As can been seen from eq.(4),(5),(7),(8) the different parameterizations focus on specific types of porous media and relate different microstructural properties to permeability. While properties such as porosity and mean grain diameter are relatively straightforward to determine, others, such as specific surface and tortuosity, are much harder to access. This is why several parameterizations have been developed to quantify these properties (Comiti and Renaud, 1989; Pech, 1984; Mota et al., 2001; Pape et al., 2005). These studies either use experimental, analytical or numerical approaches for mostly two dimensional porous media with porosities $> 30\%$.

Since the ascent of Digital Rock Physics (DRP), it has become viable to study microstructures of porous media in more detail using micro Computed Tomography (micro-CT) and Nuclear Magnetic Resonance (NMR) images (Arns et al., 2001; Arns, 2004; Dvorkin et al., 2011). Together with numerical models, these images can then be used to compute fluid flow within

porous media to determine their permeability. For this purpose several numerical methods including Finite Elements (FEM), Finite Differences (FDM) and Lattice Boltzmann method (LBM) (Saxena et al., 2017; Andrä et al., 2013a; Gerke et al., 2018; Shabro et al., 2014; Manwart et al., 2002; Bird et al., 2014) have been used.

Yet, very few data sets exist that systematically investigate microstucture (porosity and specific surface) and related flow parameters (tortuosity and permeability), in particular at porosities $< 30\%$. Most of the previous studies either measure permeability experimentally without investigating its microstructure or compute permeability and related microstructural parameters, that cannot be compared to experimental data sets. To remedy this issue, we here sinter porous glass bead samples with porosities ranging from $1.5\% - 21\%$ and investigate their microstructure using image processing. This porosity range is representative of sedimentary rocks up to a depth of $\approx 20$ km (Bekins and Dreiss, 1992). Permeability is then measured experimentally using a permeameter (see sec.2.2; Takeuchi et al. (2008); Okumura et al. (2009)) and numerically using the finite difference code LaMEM (see sec.2.7; Kaus et al. (2016); Eichheimer et al. (2019)). The theoretical permeability predictions described above in eqs.(4),(5),(7),(8) require microstructural input parameters such as porosity, specific surface and tortuosity. Within this study these parameters are determined and related to porosity. We therefore provide permeability parameterizations depending on porosity only and verify against numerically and experimentally determined values.

## 2    Methods

Here we first describe the experimental workflow including sample sintering and permeability measurement, followed by the numerical workflow featuring image processing, computation of fluid velocities and determination of both tortuosity and permeability. Fig.1 shows an overview of the entire workflow which will be explained in detail in the following section.

### 2.1    Sample sintering

Glass bead cylinders with different porosities were sintered under experimental conditions as summarized in Table 1. For this purpose soda-lime glass beads with diameters ranging from $0.9$ to $1.4$ mm were utilized as starting material (see grain size distribution in appendix D). For each sample, we prepared a graphite cylinder with $8.0$ mm inner diameter and $\approx 10$ mm height. Additional samples with diameters of $10$ and $14$ mm were prepared to check for size effects (see tab.1a). At the bottom of the graphite cylinder a graphite disc ($11.5$ mm diameter and $3.0$ mm thick) was attached using a cyanoacrylate adhesive (see fig.2 inset). The glass beads were poured into the graphite cylinder and compressed with steel rods (8-14 mm diameter) before heating.

The glass bead samples were then sintered in a muffle furnace at $710\,^{\circ}$C under atmospheric pressure. The temperature of $710\,^{\circ}$C was found to be suitable for sintering of the glass beads as it is slightly below the softening temperature of soda-lime glass around $720 - 730\,^{\circ}$C (Napolitano and Hawkins, 1964) and well above the glass transition temperature of soda-lime glass at $\approx 550\,^{\circ}$C (Wadsworth et al., 2014). At $710\,^{\circ}$C the viscosity of the employed soda-lime glass is on the order of $10^7\,\mathrm{Pas}$ (Kuczynski, 1949; Napolitano and Hawkins, 1964; Wadsworth et al., 2014) allowing for viscous flow of the glass beads at their

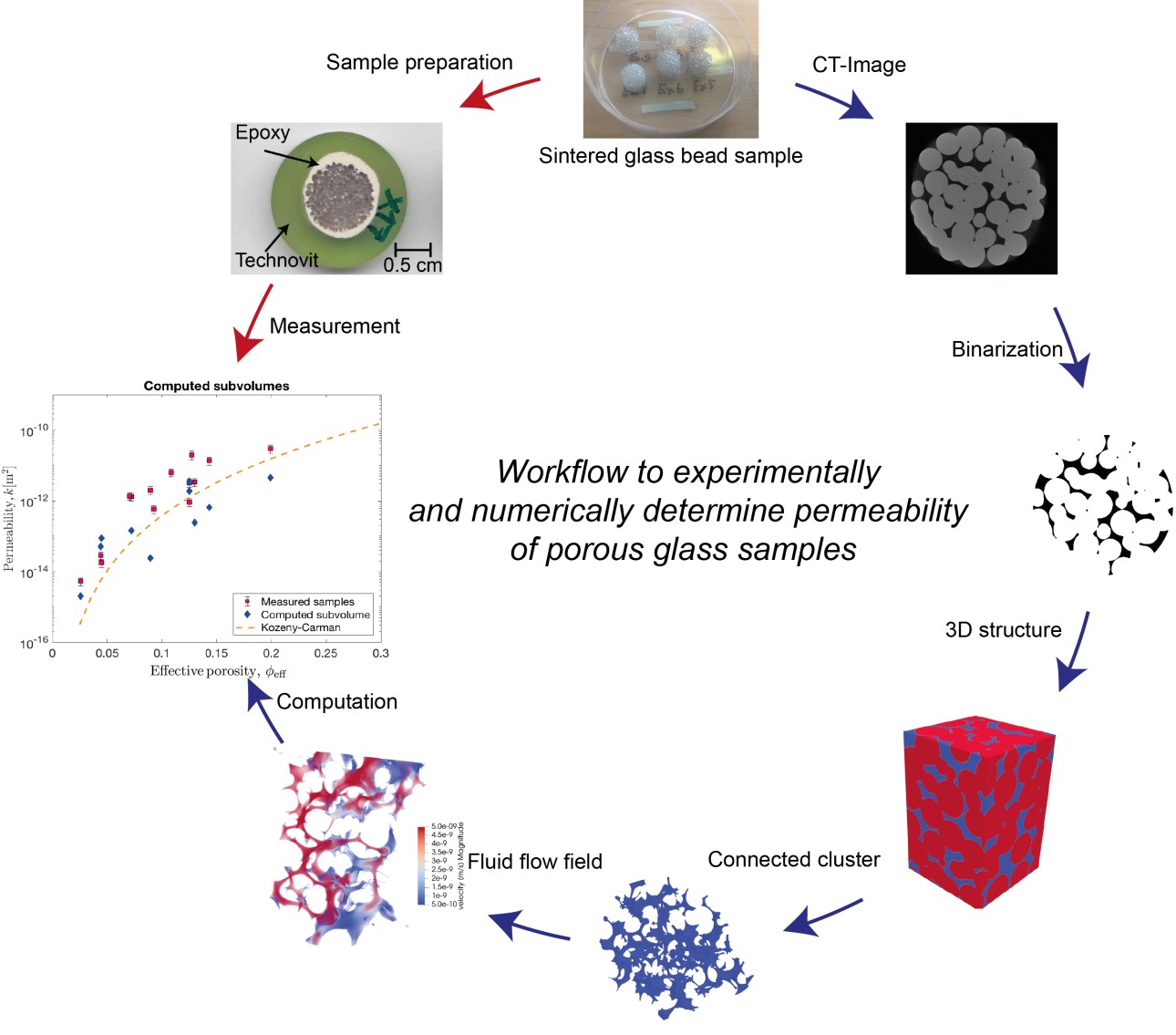

**Figure 1.** Workflow process map - red arrows mark the experimental workflow, whereas blue arrows indicate the numerical workflow.

contact surface driven by surface tension. Using different time spans ranging from $60 - 600$ minutes the viscous flow at $710\,^\circ$C controls the resulting porosity of the sample.

After sintering, the sample was cooled down to $550 - 600\,^\circ$C within $\approx$5 minutes. Afterwards the sample was taken out of the furnace to adjust to room temperature and prevent thermal cracking of the sample. In a next step the graphite container was removed from the sample. It should be noted that during the process of sintering gravity slightly affects the porosity distribution

120

within the glass bead sample (see fig.2). However, the subsamples used to compute the numerical permeability do not cover the whole height of the sample, thus the effect of compaction on the results is limited.

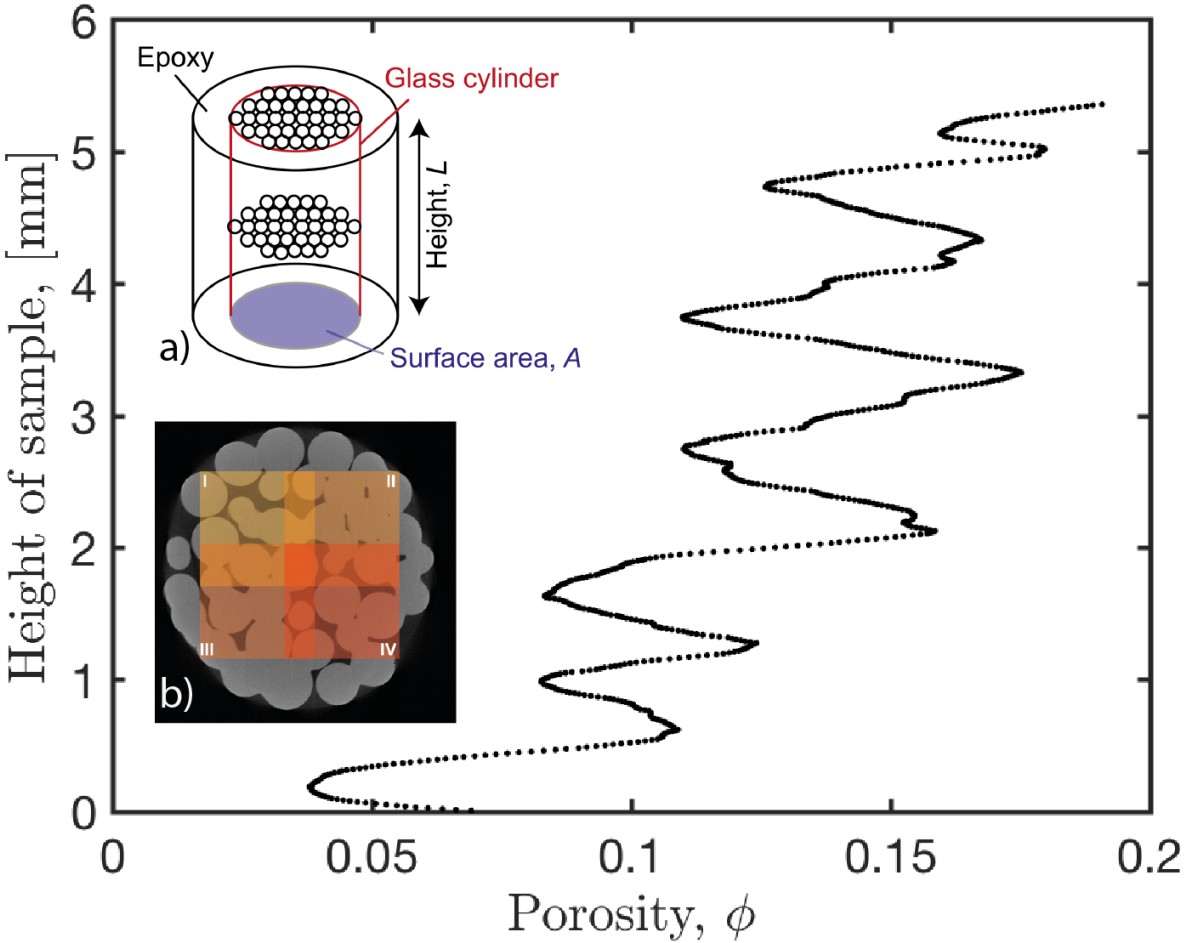

**Figure 2.** Computed porosity of each CT-slice from top to the bottom of a full sample (z-axis; sample Ex14). The diagram shows that gravity affects the porosity of the sample. Porosity minima correspond to distinct layers of glass bead within the sample. The inset **a)** provides a sketch of the sample structure. In the inset the red color outlines the cylindrical shape, blue the surface area $A$ of the cylinder and $L$ the height of the sample. **b)** shows chosen locations for the squared subsamples 1-4. Additional four subsamples (5-8) are placed similarly below subsamples 1-4 overlapping in z-direction.

## 2.2 Experimental permeability measurement

In a first preparation step we wrap a highly viscous commercial water resistant resin around the sample to avoid pore space infiltration. In a next step we embed the sample within a less viscous resin (Technovit 4071, Heraeus Kulzer GmbH & Co.

125

or Presin, Nichika Inc.) to create an airproof casing. The upper and lower surface of the sample were grinded and polished to prevent leaks during experimental permeability measurements (fig.1; Sample preparation).

The experimental permeability measurements were conducted at Tohoku University using a permeameter, described in Takeuchi et al. (2008) and Okumura et al. (2009). To determine the permeability the air flow through a sample is measured at room temperature. The pressure gradient between sample inlet and outlet is controlled by a pressure regulator (RP1000-8-04, CKD Co.; Precision $\pm 0.1\%$) at the inlet side. To monitor the pressure difference a digital manometer (testo526-s, Testo Inc.; Precision $\pm 0.05\%$) is used. Air flow through the sample is measured using a digital flow meter (Alicat, M-10SCCM; Precision $\pm 0.6\%$). As Darcy's law assumes a linear relationship between the pressure and flow rate, we measure the gas flow rate at several pressure gradients (see fig.C1 in Appendix C) to verify our assumption of laminar flow conditions. The permeability of all samples is calculated using Darcy's law (eq.(1)) based on measured values (tab.1a).

**Table 1. a)** displays experimental parameters of sintering conditions and parameters used to compute permeability using Darcy's law. $A$ denotes the sample surface area, $L$ the height of the glass bead cylinders and $D$ the inner diameter of each capsule. Additionally, the sintering time $t_{\mathrm{sint}}$, the total weight of the glass beads $m$, and the experimentally measured permeability $K_{\mathrm{meas}}$ are given. In **b)**, we list the total, effective and minimum effective porosity $\phi_{\mathrm{tot}}$, $\phi_{\mathrm{eff}}$, $\min(\phi_{\mathrm{eff}})$ of each sample. These porosities have been obtained with image processing (see sec. 2.4).

| | a) Experimental parameters | | | | | | b) Numerical parameters | | |
|---|---|---|---|---|---|---|---|---|---|
| Sample | Area | Height | Capsule Ø | Time | Tot. weight | Permeability | Porosity | Porosity | Porosity |
| | $A$ | $L$ | $D$ | $t_{\mathrm{sint}}$ | $m$ | $K_{\mathrm{meas}}$ | $\phi_{\mathrm{tot}}$ | $\phi_{\mathrm{eff}}$ | $min(\phi_{\mathrm{eff}})$ |
| | (cm$^2$) | (mm) | (mm) | (min) | (g) | (m$^2$) | (%) | (%) | (%) |
| X02 | 0.438 | 5.11 | 8 | 120 | 0.574 | $(3.1 \pm 0.2) \times 10^{-11}$ | 20.94 | 20.94 | 11.38 |
| X11 | 0.434 | 3.63 | 8 | 180 | 0.575 | $(1.91 \pm 0.09) \times 10^{-14}$ | 6.72 | 4.75 | 1.80 |
| X14 | 0.407 | 5.12 | 8 | 60 | 0.576 | $(3.4 \pm 0.2) \times 10^{-12}$ | 13.28 | 13.22 | 4.26 |
| X15 | 0.412 | 4.76 | 8 | 480 | 0.575 | $(5.7 \pm 0.3) \times 10^{-15}$ | 2.54 | 1.21 | 0.96 |
| X16 | 0.808 | 5.05 | 10 | 120 | 0.899 | $(3.1 \pm 0.2) \times 10^{-14}$ | 6.07 | 4.50 | 2.66 |
| X17 | 1.569 | 5.18 | 14 | 120 | 1.762 | $(1.41 \pm 0.07) \times 10^{-12}$ | 12.90 | 12.85 | 10.77 |
| X29 | 0.441 | 4.55 | 8 | 300 | 0.576 | $(6.3 \pm 0.3) \times 10^{-13}$ | 9.01 | 8.97 | 5.95 |
| X30 | 0.420 | 4.81 | 8 | 600 | 0.574 | $(1.52 \pm 0.08) \times 10^{-12}$ | 7.12 | 7.03 | 4.18 |
| X31 | 0.423 | 4.73 | 8 | 300 | 0.576 | $(2.1 \pm 0.1) \times 10^{-12}$ | 9.92 | 9.87 | 6.12 |
| X32 | 0.342 | 4.47 | 8 | 480 | 0.576 | $(3.7 \pm 0.2) \times 10^{-12}$ | 13.52 | 13.44 | 8.93 |
| X33 | 0.412 | 4.80 | 8 | 180 | 0.575 | $(1.53 \pm 0.08) \times 10^{-11}$ | 15.97 | 15.96 | 11.33 |
| X35 | 0.411 | 4.78 | 8 | 360 | 0.575 | $(2.2 \pm 0.1) \times 10^{-11}$ | 14.17 | 14.15 | 8.92 |
| X36 | 0.372 | 4.15 | 8 | 420 | 0.575 | $(6.9 \pm 0.4) \times 10^{-12}$ | 10.71 | 10.67 | 6.78 |

### 2.3 Micro-CT images and segmentation

Before preparing the samples for permeability measurements all samples are digitized using micro Computed Tomographic scans (micro-CT) performed at Tohoku University (ScanXmate-D180RSS270) with a resolution $\approx 6-10\,\mu$m according to the method of Okumura and Sasaki (2014). Andrä et al. (2013b) showed that the process of segmentation of the micro-CT images may have a significant effect on the three dimensional pore space and therefore the computed flow field. In two-phase systems (fluid + mineral), as in this study, the segmentation is straightforward due to the high contrast in absorption coefficients between glass beads and air, while it can become quite complex for multiphase systems featuring several mineral phases. In the present study the segmentation of the obtained micro-CT images was done using build-in MatLab functions. In a first step the images are binarized using Otsu's method (Otsu, 1979). Additional smoothing steps of the images are performed. In a next step the two dimensional micro-CT slices are stacked on top of each other, resulting in a three dimensional representation of the pore space (fig.1; 3D structure).

### 2.4 Porosity determination

Porosity is an important parameter describing microstructures. It is defined as the ratio of the total pore space $V_V$ to the bulk volume of the sample $V_b$ (Bird et al., 2006):

$$\phi_{\mathrm{tot}} = \frac{V_V}{V_b} \tag{9}$$

In a first step, the total porosity of each sample is determined by counting the amount of solid and fluid voxels. In a second step, we determine the isolated pore space using a flooding algorithm implemented in MatLab (bwconncomp). This isolated pore space is then subtracted from the total pore space to obtain an effective pore space $V_{\mathrm{eff}}$. As a bonus, this procedure reduces the computational cost for numerical permeability determinations by removing the parts of the pore space that do not contribute to fluid flow and thus permeability. The effective porosity $\phi_{\mathrm{eff}}$ is then defined as the volume of all percolating pore space clusters $V_{V_{\mathrm{eff}}}$ to the bulk volume of the sample:

$$\phi_{\mathrm{eff}} = \frac{V_{V_{\mathrm{eff}}}}{V_b} \tag{10}$$

It should be mentioned that in a simple capillary model $\phi_{\mathrm{eff}} = \phi$ since no isolated pore space exists. It should also be noted that only the effective porosity is used to determine microstructural and flow properties later in this study.

As described in section 2.1, the porosity of the samples is not homogeneous, but increases towards the sample bottom due to gravity. As permeability may not necessarily be affected by the total porosity, but rather by the minimum effective porosity in a sample (in a slice perpendicular to the flow direction), we also determined the minimum effective porosity of each sample (see tab.1b).

## 2.5 Effective specific surface

The specific surface is defined as the total interfacial surface area of pores $A_s$ per unit bulk volume $V_b$ of the porous medium (Bear, 1988):

$$S = \frac{A_s}{V_b} \tag{11}$$

As in the previous section we compute the effective specific surface of all percolating pore space clusters and neglect isolated pore space. To determine the effective specific surface we use the extracted connected clusters and compute an isosurface of the entire three dimensional binary matrix. In a next step the area of the resulting isosurface $A_s$ is calculated.

## 2.6 Numerical method

The relationship between inertial and viscous forces in fluid flows is described by the Reynolds number:

$$Re = \frac{\rho v L}{\eta}, \tag{12}$$

where $\rho$ is the density, $v$ the velocity component, $L$ denotes the length of the domain and $\eta$ is the viscosity of the fluid. For laminar flow conditions ($Re < 1$, see fig.C1 Appendix C) and ignoring gravity, the flow in porous media can be described with the incompressible Stokes equations:

$$\frac{\partial v_i}{\partial x_i} = 0 \tag{13}$$

$$\frac{\partial}{\partial x_j}\left[\eta\left(\frac{\partial v_i}{\partial x_j} + \frac{\partial v_j}{\partial x_i}\right)\right] - \frac{\partial P}{\partial x_i} = 0 \tag{14}$$

with $P$ being the pressure and $x$ the spatial coordinate. For all simulations, we employed a fluid viscosity of 1 Pas.

The Stokes equations are solved using the finite difference code LaMEM (Kaus et al., 2016; Eichheimer et al., 2019). LaMEM employs a staggered grid Finite Difference scheme (Harlow and Welch, 1965), where pressures $P$ are defined at the cell centers and velocities $v$ at cell faces. Based on the data from the CT-scans, each cell is assigned either a fluid or a solid phase. The discretized system of equations is then solved using multigrid solvers of the PETSc library (Balay et al., 2019). As only cells within the fluid phase contribute to fluid flow the discretized governing equations are only solved for these cells. This greatly decreases the number of degrees of freedom and therefore significantly reduces the computational cost. Due to computational limitations and the densification at the bottom of the samples (see fig.2) we extract 8 overlapping subvolumes per full sample (see fig.2b), with sizes of $512^3$ cells. For each subvolume we compute effective porosity, effective specific surface, hydraulic tortuosity and permeability.

## 2.7 Numerical permeability computation

From the calculated velocity field in $z$-direction the volume-averaged velocity component $v_m$ is calculated (e.g. Osorno et al., 2015):

$$v_m = \frac{1}{V_f} \int_{V_f} |v_z| \, dv, \tag{15}$$

where $V_f$ is the volume of the fluid phase. Using Darcy's law (eq.1; Andrä et al., 2013a; Bosl et al., 1998; Morais et al., 2009; Saxena et al., 2017) an intrinsic permeability $k_s$ is computed via:

$$k_s = \frac{\eta v_m}{\Delta P} \tag{16}$$

## 2.8 Hydraulic tortuosity

Tortuosity is not only highly relevant for the Kozeny-Carman relation, but is also used in various engineering and science applications (Nemati et al., 2020). It has a major influence on liquid-phase mass transport (e.g. in Li-ion batteries (Thorat et al., 2009) and membranes (Manickam et al., 2014)), the effectiveness of tertiary oil recovery (Azar et al., 2008) and evaporation of water in soils (Hernández-López et al., 2014). In recent years, several definitions for tortuosity have been suggested (Clennell, 1997; Bear, 1988; Ghanbarian et al., 2013). For the remainder of this study we will calculate and apply the so-called hydraulic tortuosity (Ghanbarian et al., 2013). Assuming that hydraulic tortuosity changes with porosity, both numerical and experimental studies published different relations of hydraulic tortuosity to porosity. In most of the cases the hydraulic tortuosity is assumed to be constant as it is difficult to determine experimentally, which is rarely done. It should be mentioned that the following hydraulic tortuosity-porosity relations have been obtained for porous media with $> 30\%$ porosity.

Matyka et al. (2008) numerically determined the hydraulic tortuosity by using an arithmetic mean given as:

$$\tau_h = \frac{1}{N} \sum_{i=1}^{N} \tau(r_i), \tag{17}$$

where $\tau = L_e/L$ is the hydraulic tortuosity of a flow line crossing through point $r_i$ (eq.(3)) and $N$ the total number of streamlines.

Koponen et al. (1996) computed the hydraulic tortuosity numerically using:

$$\tau_h = \frac{\sum_i \tau^n(r_i) v(r_i)}{\sum_i v(r_i)}, \tag{18}$$

where $v(r_i) = |v(r_i)|$ is the fluid velocity at point $r_i$ and points $r_i$ are chosen randomly from the pore space (Koponen et al., 1996).

One of the most common relations for hydraulic tortuosity is a logarithmic function of porosity reading as follows:

$$\tau_h(\phi) = 1 - B ln(\phi), \tag{19}$$

where $B$ is a constant found experimentally for different particles (e.g. 1.6 for wood chips (Pech, 1984; Comiti and Renaud, 1989), 0.86 to 3.2 for plates (Comiti and Renaud, 1989)). By numerically computing hydraulic tortuosity for two dimensional squares, Matyka et al. (2008) obtained $B = 0.77$. A different experimental relation for hydraulic tortuosity measuring the electric conductivity of spherical particles was proposed by Mota et al. (2001):

$$\tau_h(\phi) = \phi^{-0.4} \tag{20}$$

Investigating numerically two-dimensional porous media with rectangular shaped particles Koponen et al. (1996) proposed a different relation:

$$\tau_h(\phi) = 1 + 0.8(1 - \phi) \tag{21}$$

In the present study the hydraulic tortuosity is determined according to eq.(17), which requires to compute the tortuosity $\tau$ of individual streamlines within each sample. Streamlines describe a curve traced out in time by a fluid particle with fixed mass and are described mathematically as:

$$\frac{\partial x_i}{\partial t} = v(x, t), \tag{22}$$

with $v$ being the computed velocity field obtained from the numerical simulation and $t$ being the time. Integrating eq.(22) yields

$$x_i = x_i(x^0, t), \tag{23}$$

where $x^0$ is the position of the prescribed particle at $t = 0$. Eq.(22) is solved using built-in MatLab ODE (Ordinary Differential Equation) solvers. To compute the streamline length all fluid cells at the inlet of the subsample are extracted and used as streamline starting points. Using the computed velocity field and eq.(22) the streamline length for each starting point is calculated. Hence, up to 40000 streamlines need to be computed for a subsample with $\approx 20\%$ porosity, whereas for a subsample with $\approx 5\%$ porosity up to 5000 streamlines are computed.

## 3   Results

In this section we analyze the different samples in terms of porosity, specific surface, hydraulic tortuosity and permeability. All data for each subsample presented here are given in the supplementary tables (see table 1 - 13). Effective porosity and effective specific surface are computed for both subsamples and full samples, whereas hydraulic tortuosities and permeabilities are only computed for subsamples due to computational limitations. In the present study we analysed 13 samples and 104 subsamples.

### 3.1   Porosity

The total porosity for each sample and subsample is analysed using image processing and ranges from $2.5 - 21\%$ (see tab. 1b and supplement table 1-13). The effective porosity is determined by extracting all connected clusters within the samples and

ranges from $1.21 - 21\%$ (see also tab.1b). The analysis of the micro CT images also showed that during sintering densification

of the samples occurs (see fig.2). For this reason we furthermore report the minimum effective porosity $min(\phi_{\text{eff}})$. Assuming an effective porosity for the entire sample therefore does not seem to be representative as during the laboratory measurements a first order control mechanism of the fluid flow and therefore permeability is the lowest porosity within the entire sample.

## 3.2 Effective specific surface

Figure 3 shows the computed specific surfaces for all subsamples and all full samples with increasing effective porosity.

Koponen et al. (1997) used the following relationship to predict the specific surface:

$$S = -\frac{n}{R_0}\phi_{\text{eff}}ln(\phi_{\text{eff}}), \tag{24}$$

where $n$ is the dimensionality and $R_0$ is the hydraulic radius of the particles. The hydraulic radius is defined as $2V_p/M$ (e.g. Bernabé et al., 2010), with $V_p$ being the pore volume and $M$ being the pore surface area. For a regular simple cubic sphere packing with $\phi = 0.476$ the estimated hydraulic radius is $\approx 151\,\mu m$. To relate the computed values for the effective specific

surface to the effective porosity the above equation is fitted, resulting in a hydraulic radius of $385.09\,\mu m$:

$$S = -\frac{3}{3.8509 \times 10^{-4}\,m}\phi_{\text{eff}}ln(\phi_{\text{eff}}) \tag{25}$$

The fit between eq.(24) and our data shows good agreement which is also reflected in a value of $R^2$=0.975 (see fig.3).

## 3.3 Hydraulic tortuosity

We computed hydraulic tortuosities for all subsamples which exhibit a percolating pore space. Results are shown in fig. 4,

where we compare different hydraulic tortuosity-porosity parameterizations presented in section 2.8 to our data. In fig.4a)-c), we compare our data (denoted by grey squares) with one of the three porosity-hydraulic tortuosity parameterizations (denoted by solid and dashed lines), whereas in fig.4d), we show a simple linear fit to our data. In general, computed hydraulic tortuosities are quite scattered and show variations ranging from values of about 2 to values of around 4. In fig.4a) we compare our data to the hydraulic tortuosity parameterization from Matyka et al. (2008) (see eq.(19)), which is denoted by a dashed black

line. We refitted this parameterization using our data, with the result shown by the red solid line with corresponding 95% confidence bounds with the coefficient of determination $R^2 = -1.6317$. In fig.4b) and c), similar comparisons are shown, but for the parameterizations by Koponen et al. (1996) (fig.4b) and Mota et al. (2001) (fig.4c). In both cases, we show the original parameterizations as a black dashed line and the fitted parameterizations as a colored solid line with colored dashed lines indicating the 95% confidence bounds. As for the parameterization by Matyka et al. (2008), these two parameterizations do

not fit our data very well, as is also indicated by their low $R^2$ values ($R^2 = -5.6017$ and $R^2 = 0.0758$ respectively). Finally, in fig.4c), we show a linear fit to our data together with the 95% confidence bounds. As indicated by the low $R^2$ value of 0.0274, this fit does also not represent the data very well. For this reason we use the arithmetic mean of the computed hydraulic tortuosities for later permeability predictions. Nevertheless, we do observe that despite the large scatter, hydraulic tortuosity largely remains relatively constant with decreasing porosity, thus indicating that the pore distribution of our experimental

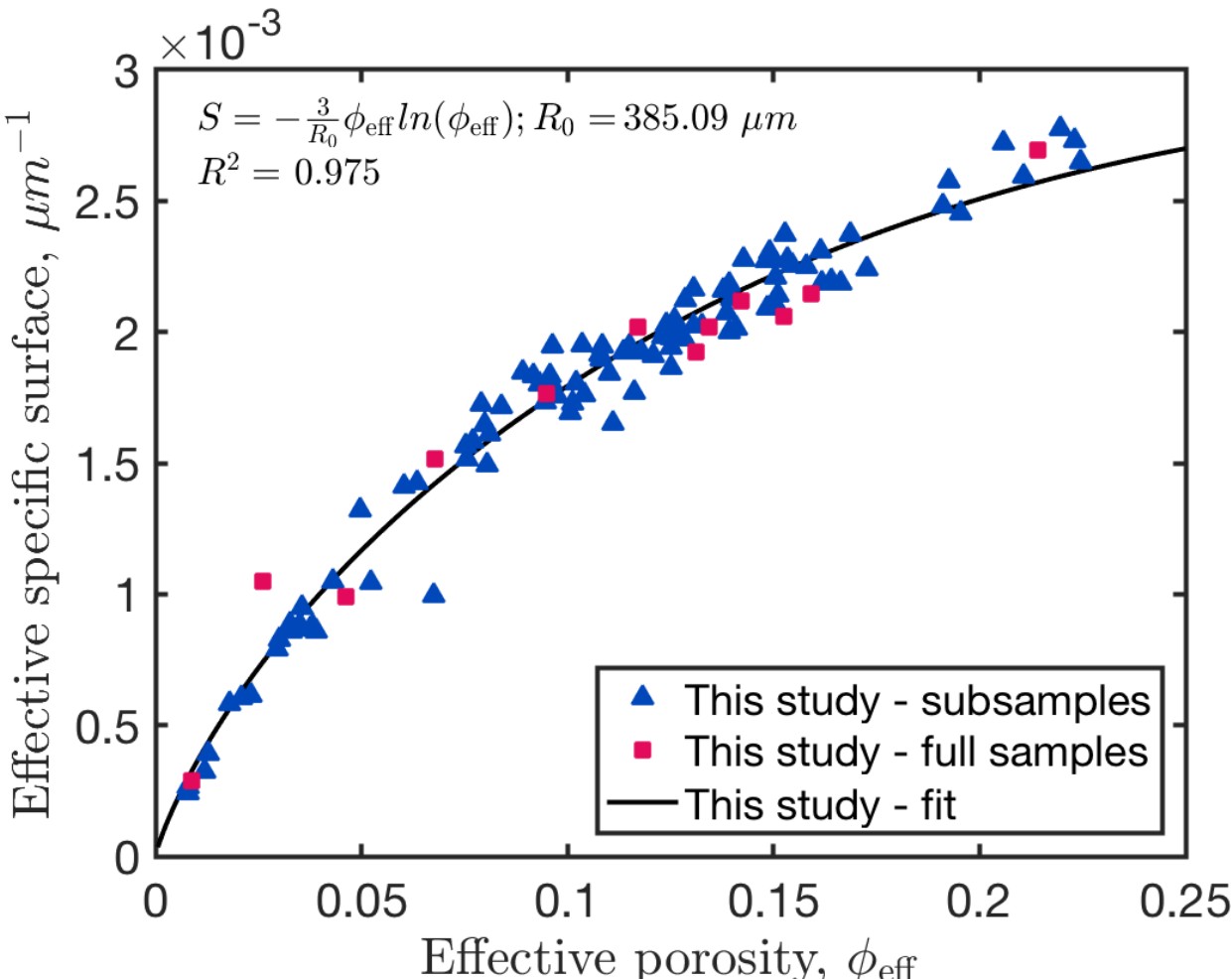

**Figure 3.** Effective specific surface as a function of effective porosity. Blue triangles represent subsample data from this study and red squares the effective specific surface of full samples. Full sample data points are plotted in order to show that in terms of effective specific surface subsamples represent full samples very well. The black curve represents the fitted curve according to eq.(25).

products is homogeneous and the geometrical similarity of pore structure was kept during sintering. This is in contrast to the parameterizations of Matyka et al. (2008) and Mota et al. (2001), both predicting a significant increase in tortuosity as small porosities are approached, but agrees with the model of Koponen et al. (1996).

### 3.4  Permeability

In fig.5, measured permeabilities for all samples are shown as grey symbols (see also tab.1a for measured values). We here
chose to plot sample permeabilities vs. the minimum of the effective porosity, the reason being the intrinsic porosity variations

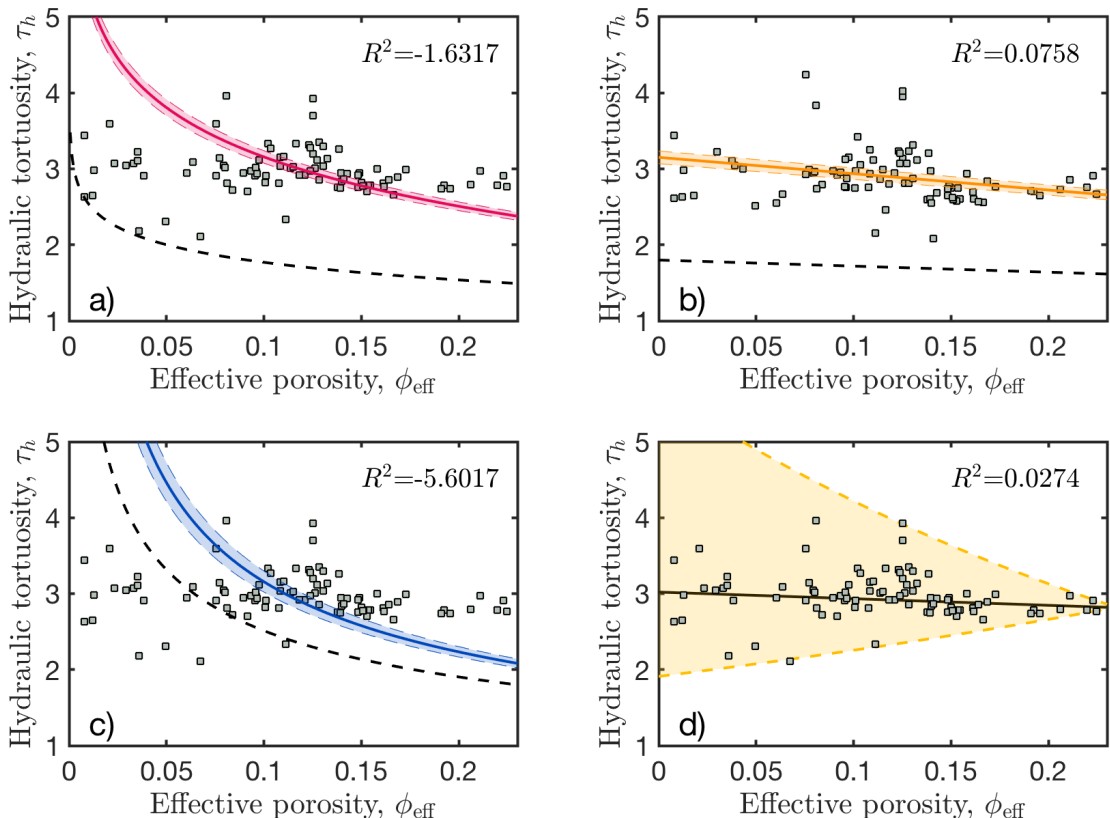

**Figure 4. (a)-(c)** show the proposed relations for the hydraulic tortuosity according to **(a)** Matyka et al. (2008), **(b)** Koponen et al. (1996) and **(c)** Mota et al. (2001) as black dashed lines. The colored solid lines represent the fit of the computed data to those relations within the 95% confidence bounds. Hydraulic tortuosities for all subsamples (grey squares) are computed according to the method used in each of these studies. **(d)** shows the fit obtained in the present study. The colored area in **(d)** illustrates the extending distribution of computed hydraulic tortuosities with decreasing effective porosity.

in each sample (see section 2.4). Figure A1 in the appendix shows both the effective porosity and minimum effective porosity of each sample.

Measured permeabilites range from values of around $10^{-14}$ m$^2$ to about $10^{-11}$ m$^2$, depending on porosity. Although experimental measurements are scattered, a clear trend can be observed. At porosities close to the critical porosity, permeabilities are very low, but rapidly increase when porosities increase slightly. At larger porosities, permeabilities further increase, but this increase is significantly less rapid.

Numerically 98 subsamples have been computed successfully with permeabilities ranging from around $10^{-14}$ m$^2$ to about $10^{-10}$ m$^2$, depending on porosity (see table 1-13 in the supplement and fig.B1 in the appendix). In comparison to the experimentally measured samples, the numerical permeabilities tend towards higher values, but show a clear trend.

As we split each sample in eight subsamples for numerical permeability computations, we need to average them to compute an effective sample permeability that can then be compared to measured values. This upscaling issue is not trivial to address and it is not clear yet which averaging method is appropriate. It is possible to put bounds on the effective permeability by using either the arithmetic or harmonic mean of subsample permeabilities. However, these bounds correspond to very specific geometrical sample structures. In the case of the arithmetic mean, the medium is assumed to consist of parallel layers oriented parallel to the flow direction whereas the harmonic mean is valid in the case of parallel layers orthogonal to the flow direction. This is most often not the case. Therefore, different averaging methods have been developed to obtain adequate upscaling procedures for heterogeneous porous media (e.g. Sahimi, 2006; Jang et al., 2011; Torquato, 2013). One of the simplest averaging schemes that has been shown to be an appropriate approximation for heterogeneous porous media is the geometric mean (e.g. Warren and Price, 1961; Selvadurai and Selvadurai, 2014; Jang et al., 2011), which reads as:

$$k_g = \left( \prod_{i=1}^{n} k_i \right)^{1/n} \tag{26}$$

where $i$ is the number of the subsample and $n$ the total number of subsamples (eight in this study). As several subsamples at low porosities did not exhibit a connected pore space (thus not allowing for any fluid flow), we assumed a permeability of $10^{-20}$ m$^2$ for these samples. The geometric averages of each subsample set are shown in fig.5.

To determine the predictive power of the different permeability parameterizations described in section 1, we inserted the expressions for effective specific surface and hydraulic tortuosity into the respective equations (eq.(4) & (5)).

The Kozeny-Carman equation then reads as:

$$k = k_0 \frac{[min(\phi_{\text{eff}}) - \phi_{\text{c}}]^3}{2.9715^2 \cdot \left[ -\frac{3}{3.8509 \times 10^{-4} \, m} \phi_{\text{eff}} ln(\phi_{\text{eff}}) \right]^2}, \tag{27}$$

with $k_0 = 0.5$ being the geometrical parameter for spherical particles (Kozeny, 1927) and $\phi_c = 0.01$ as the critical porosity threshold. This threshold is lower than the published value of $\phi_c = 0.03$ (Van der Marck, 1996; Rintoul, 2000; Wadsworth et al., 2016). However, one of the subsamples used in this study had a porosity of 0.01 while still exhibiting a percolating cluster. For this reason, we here employed a critical porosity of $\phi_c = 0.01$.

With our parameterization for $S$, the permeability parameterization of Martys et al. (1994) reads as follows:

$$k = \frac{2[1 - min(\phi_{\text{eff}}) - \phi_c]}{\left[ -\frac{3}{3.8509 \times 10^{-4} \, m} \phi_{\text{eff}} ln(\phi_{\text{eff}}) \right]^2} [min(\phi_{\text{eff}}) - \phi_c]^{4.2}, \tag{28}$$

From the grain size distribution of the glass beads used in this study (see Appendix D), we also determined the average grain diameter $d$ and the harmonic mean diameter $D$, both within uncertainties equal to 1.20 mm. Inserting into the respective parameterizations of Revil and Cathles III (1999) and Garcia et al. (2009) (see (7) and eq.(8)) results in:

$$k = \frac{[1.20 \times 10^{-3} \, m]^2 min(\phi_{\text{eff}})^{5.1}}{24}, \tag{29}$$

$$k = min(\phi_{\text{eff}})^{0.11}[1.20 \times 10^{-3}\,m]^2 \tag{30}$$

The permeability parameterizations in general show similar trends but differ in the predicted permeability value. The Kozeny-Carman relation shows good agreement with the experimentally measured samples, but also shows some offset towards the numerically computed values. A similarly good fit is obtained by the permeability parameterization of Martys et al. (1994). The parameterizations by Garcia et al. (2009) and Revil and Cathles III (1999) tend to underestimate permeability, which might be related to their assumptions on the samples heterogeneity.

## 4  Discussion and conclusion

In this paper, we determine the permeability of nearly isotropic porous media consisting of sintered glass beads using a combined experimental-numerical approach. We analyzed sample microstructures using CT data and determined flow properties both experimentally and numerically. Using this data, we test different permeability parameterizations that have been proposed in the literature. The goal of this study was to particularly improve permeability parameterizations at low porosities (<20%).

Two particular microstructural parameters that we determined were the specific surface $S$ and the hydraulic tortuosity $\tau_h$. As these two parameters are frequently used in permeability parameterizations, we tested whether existing parameterizations are also valid in our case. We find that the effective specific surface is well predicted by the parameterization eq.(24) proposed by Koponen et al. (1996), not only for the chosen subsamples but also for the full samples. The fitted hydraulic radius of $0.385\,\mathrm{mm}$ is reasonable as the initial grain size of the glass beads is around $1\,\mathrm{mm}$ and the hydraulic pore radius of the glass beads is reduced during sample sintering.

Only few studies have investigated hydraulic tortuosity for three dimensional porous media (Du Plessis and Masliyah, 1991; Ahmadi et al., 2011; Backeberg et al., 2017). As the hydraulic tortuosity is challenging to determine in experiments, experimental studies have often used this parameter as a fitting variable. Our data shows that - contrary to previous suggestions - the hydraulic tortuosity does not change significantly with decreasing effective porosity (Matyka et al., 2008; Koponen et al., 1996; Mota et al., 2001), at least at the low porosities investigated in this study. This observation agrees with the study by Koponen et al. (1996), but is at odds with the studies by Matyka et al. (2008) and Mota et al. (2001). The study by Koponen et al. (1996) was based on 2D numerical simulations and found hydraulic tortuosity values close to 2 whereas our data lies around a value of 3. The difference between previous relations and our data is likely related to the different particle geometries used and that previous studies were done in 2D, while we employ 3D samples.

Measured and computed permeabilities are generally in good agreement, with computed permeabilities consistently yielding towards higher values than experimentally measured permeabilities. The experimental measured permeabilities show some scatter which might be related to heterogeneities within the sample. Interestingly, numerical permeability computations based on subsamples show much less scatter. Both the modified Kozeny-Carman relation and the parameterization by Martys et al. (1994) predict numerically computed and experimentally measured permeability values well. In the modified Kozeny-Carman

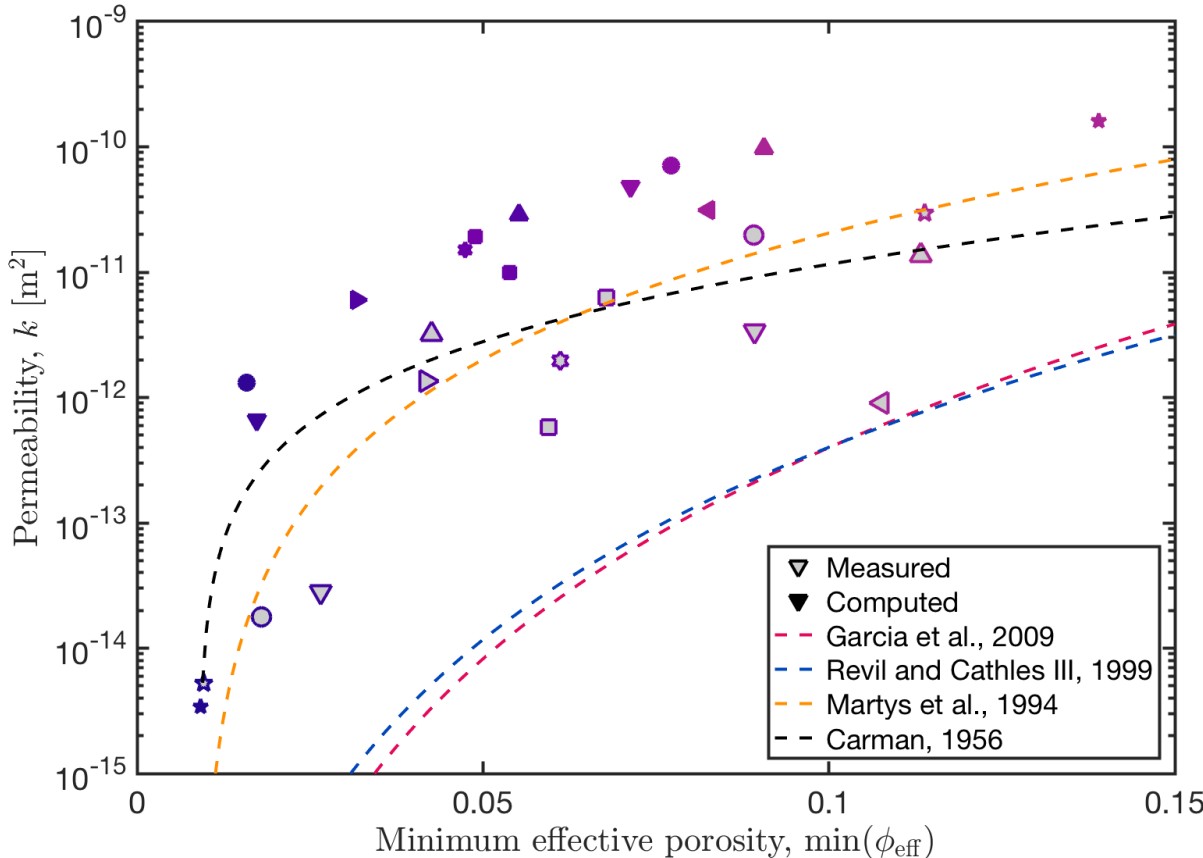

**Figure 5.** Computed and measured permeability against minimum effective porosity. Symbols of the same shape and color represent the same sample. Samples with grey face color represent measured values, whereas color only symbols stand for computed subsamples. The computed permeabilities represent the geometric mean values of all subsamples. To verify existing permeability parameterizations, we plotted the relations of Revil and Cathles III (1999); Garcia et al. (2009); Carman (1956) and Martys et al. (1994) against the experimental and numerical permeabilities. Note that estimated errors for the experimental permeability measurements (tab.1a) are smaller than the displayed symbols. Some subsamples with low effective porosity did not show a continuous pathway throughout the subsample, thus we assumed a very low permeability of $10^{-20}\,\mathrm{m^2}$.

relation, hydraulic tortuosity seems to have a second order influence on the permeability of porous media. The permeability parameterizations by Revil and Cathles III (1999) and Garcia et al. (2009) underestimate permeabilities, which could be related to the assumptions used in these studies. It should be noted that Garcia et al. (2009) investigated heterogeneous sand packs and found that permeabilities for homogeneous packs are $1.6 - 1.8$ times higher.

      There are several reasons for the discrepancy between experimental and numerical values. First, numerical permeability
predictions are based on simulations on subsamples, where free slip boundary conditions are employed. These boundary

conditions do not accurately represent the flow field within the full sample and are therefore a possible source of error. This error can be estimated to about 20-50% of the computed value (Gerke et al., 2019). Second, the numerical computations compute the flow field on a discretized grid with a given resolution. In particular at low porosities, pore structures may be too small to be well resolved by the grid. As discussed by previous studies the accuracy of permeability prediction improves with increasing numerical resolution (Gerke et al., 2018; Keehm, 2003; Eichheimer et al., 2019). To investigate this effect with respect to our samples, we computed the permeability of two subsamples (Ex35Sub04 and Ex36Sub02 see supplemental material) using an increased resolution of $1024^3$ grid points. The two samples with effective porosities at around 9 and 15% represent samples on both sides of the median of the present study's effective porosity range ($1.5 - 22\%$). The permeability obtained using doubled grid resolution decreases only by around $\approx 2 - 4\%$ compared to the outcome of models with $512^3$ grid resolution (see Appendix F). We are therefore confident that the calculations with $512^3$ grid points provide sufficiently accurate results. To further increase the accuracy of the numerical computations, adaptive meshing methods could be useful.

Third and most important, it is not clear whether either the subsamples used in the numerical computations or the full samples used for experimental measurements can be considered representative volume elements at a certain porosity. The scatter that we observe in both numerical and experimental permeability measurements indicates that this may not be the case, in particular at porosities close to the critical porosity. A potential remedy for this issue would be the use of larger samples in both experiments and numerical simulations. However, using larger samples is not trivial. On the numerical side larger samples require significantly more computational resources. On the experimental side, larger samples reduce the resolution of the CT scans, which would in turn reduce the value of microstructural analysis. Additionally, a reduced CT resolution would also affect numerical permeability measurements.

We show that several permeability parameterizations (the modified Kozeny-Carman equation and the permeability parameterization by Martys et al. (1994)) are capable to predict the numerically and experimentally determined permeabilities obtained in our study. However, this could only be done by determining several microstructural parameters from CT scans and by modifying the respective equations to fit our data. In that repsect, the parameterization by Martys et al. (1994) requires less fitting parameters, which makes it in our opinion preferable. However, our results also show a significant scatter in both numerical and experimental permeability measurements which are not predicted by either parameterization. This shows that further work is needed to obtain a more universal parameterization connecting microstructural parameters to permeability. To first order, the different permeability parameterizations can be used in numerical models to simulate fluid flow in isotropic low porosity media on the larger scale. However, it has to be kept in mind that rocks in nature are commonly more complex, as they (1) often consist of grains with different shapes and sizes, (2) contain fractures which serve as preferred pathways for fluid flow and (3) often also contain anisotropic structures.

Nevertheless, our study demonstrates that numerical permeability computations can complement laboratory measurements, in particular in cases of small sample sizes or effective porosities $< 5\%$. We provide segmented input files of several samples with different porosities in the supplementary. We hope that this will allow other researchers to use this data and our results to benchmark other numerical methods in the future.

*Code availability.* https://bitbucket.org/bkaus/lamem/src/master/ ; commit: 9c06e4077439b5492d49d03c27d3a1a5f9b65d32 (Popov and Kaus, 2016).

*Data availability.* Detailed data tables for each sample can be found in the supplementary material. The segmented CT images of three samples with different porosities are provided using the figshare repository (doi:10.6084/m9.figshare.11378517).

## Appendix A: Minimum effective porosity

This figure shows the comparison between the effective porosity and the minimum effective porosity, which may control the fluid flow within the sample. The minimum effective porosity is used in fig.5.

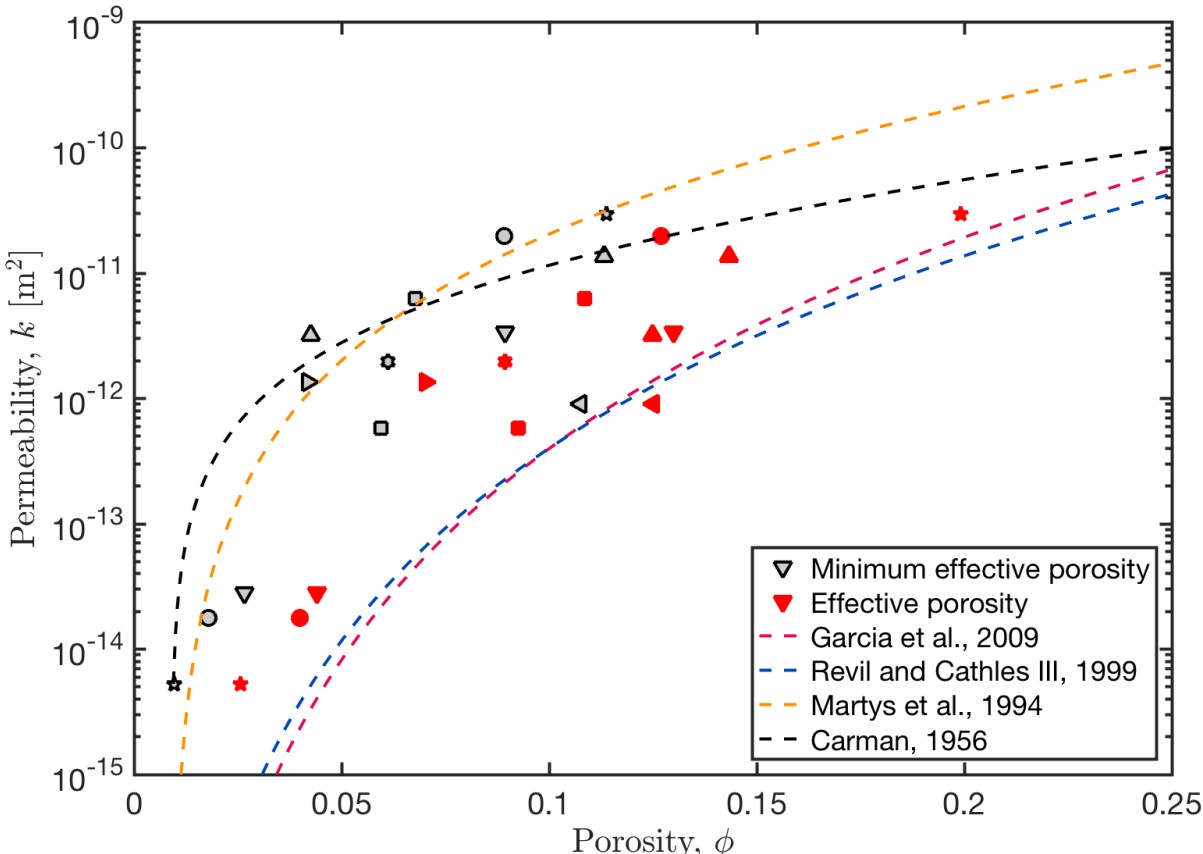

**Figure A1.** Measured permeability against porosity. Symbols with grey face color represent sample using the minimum effective porosity per sample, while red symbols display measured sample using the effective porosity. Dashed lines show several permeability parameterizations.

## Appendix B: Permability of each subsample

This figure shows the computed permeability of each subsample together with the measured permeability values and the permeability parameterizations.

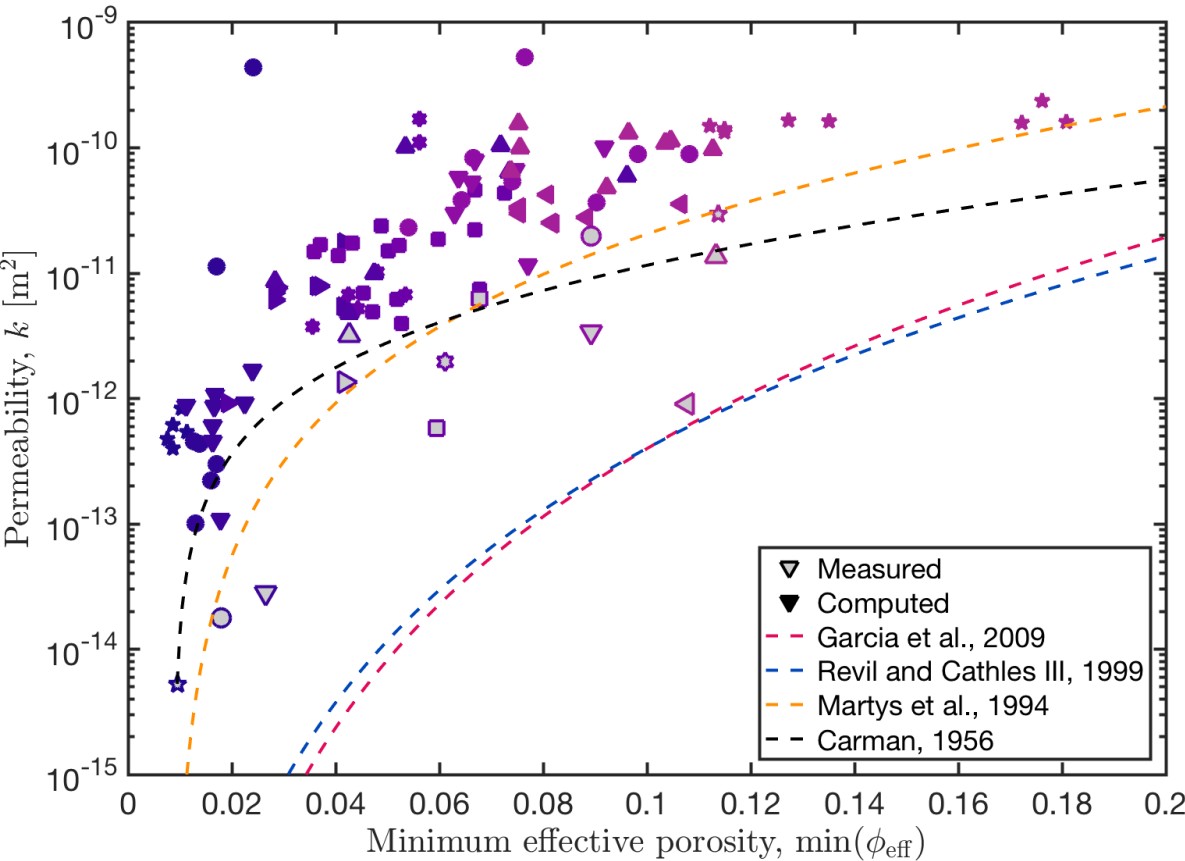

**Figure B1.** Computed and measured permeability against minimum effective porosity. Symbols of the same shape and color represent the same sample. Samples with grey face color represent measured values, whereas color only symbols stand for computed subsamples. To verify existing permeability parameterizations, we plotted the relations of Revil and Cathles III (1999); Garcia et al. (2009); Carman (1956) and Martys et al. (1994) against the experimental and numerical permeabilities. Note that estimated errors for the experimental permeability measurements (tab.1a) are smaller than the displayed symbols.

## Appendix C:  Applicability of Darcy's Law

For the numerical permeability computation using the Stokes equations we assume laminar flow conditions and incompressibility. Laminar flow conditions are represented by a linear relationship between applied pressure gradient and flow rate (fig.C1). Regarding the incompressibility of the working gas during the measurements we computed permeabilities using both Darcy's law (eq.(1)) and Darcy's law for compressible gas as follows (Takeuchi et al., 2008):

$$\frac{P_2^2 - P_1^2}{2 P_2 L} = \frac{\eta \nu_0}{k},$$ (C1)

with $P_2$ and $P_1$ being the pressures at the inlet and outlet side of the sample respectively, and $\nu_0$ being the volume flux, which is calculated from the flow rate divided by cross-sectional area of the sample. The left-hand side of Eq.(C1) represents the modified pressure gradient that includes the compressibility of working gas. The difference between both computed permeabilities is less than $10\%$, we therefore assume the effect of compressibility to be minor.

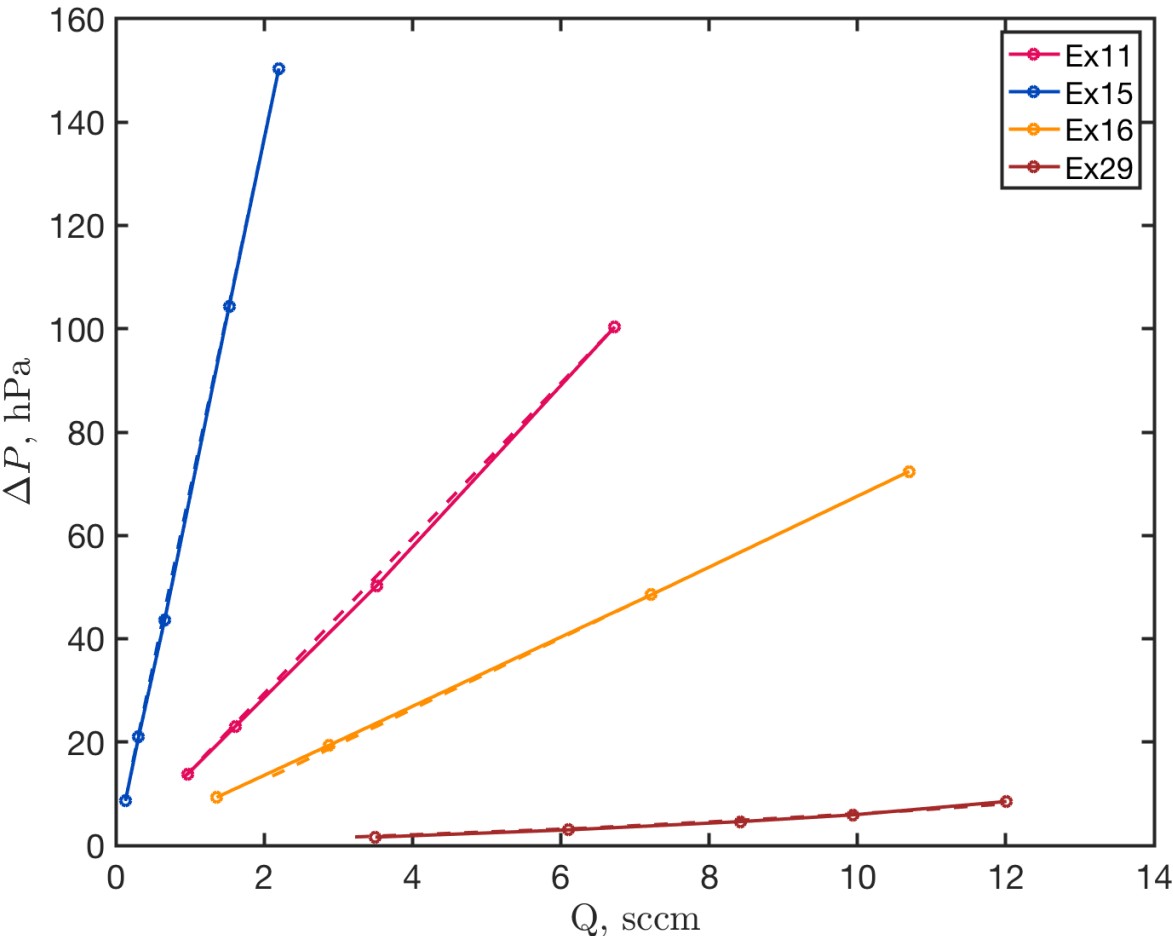

**Figure C1.** The linear relations between applied pressure difference and flow rate show that Darcy's law is valid and no turbulent flow occurs. Solid lines represent measurements while increasing the pressure difference and dashed lines while decreasing the pressure difference. The unit of *sccm* refers to a standard cubic centimeter per minute.

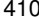

**Figure D1.** Size frequency distribution of the glass beads diameter. Beside the distribution, both arithmetic mean $\tilde{d}$ and standard deviation $\sigma$ are given.

## Appendix E: Permeability upscaling schemes

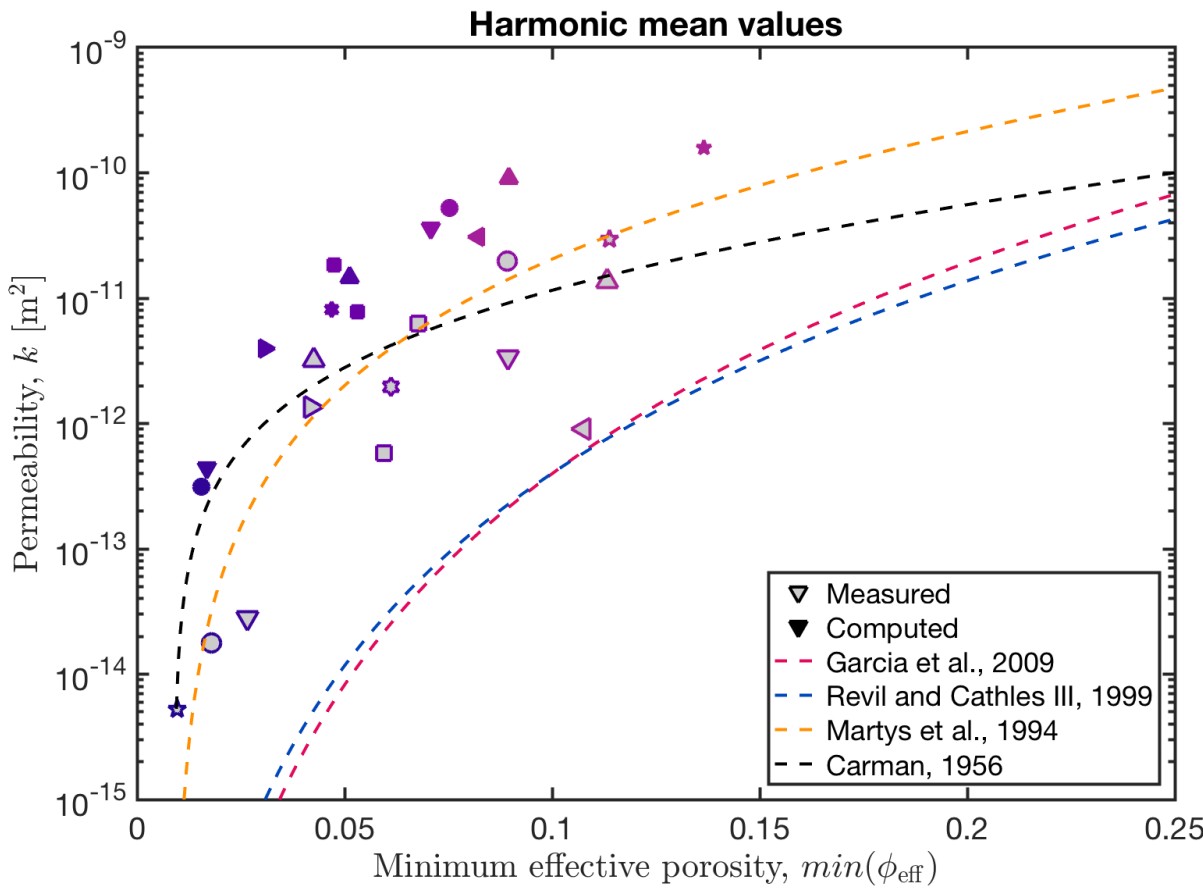

**Figure E1.** Computed and measured permeability against minimum effective porosity. Symbols of the same shape and color represent the same sample. Samples with grey face color represent measured values, whereas color only symbols stand for computed subsamples. The computed permeabilities represent the harmonic mean values of all subsamples. To verify existing permeability parameterizations, we plotted the relations of Revil and Cathles III (1999); Garcia et al. (2009); Carman (1956) and Martys et al. (1994) against the experimental and numerical permeabilities. Note that estimated errors for the experimental permeability measurements (tab.1a) are smaller than the displayed symbols.

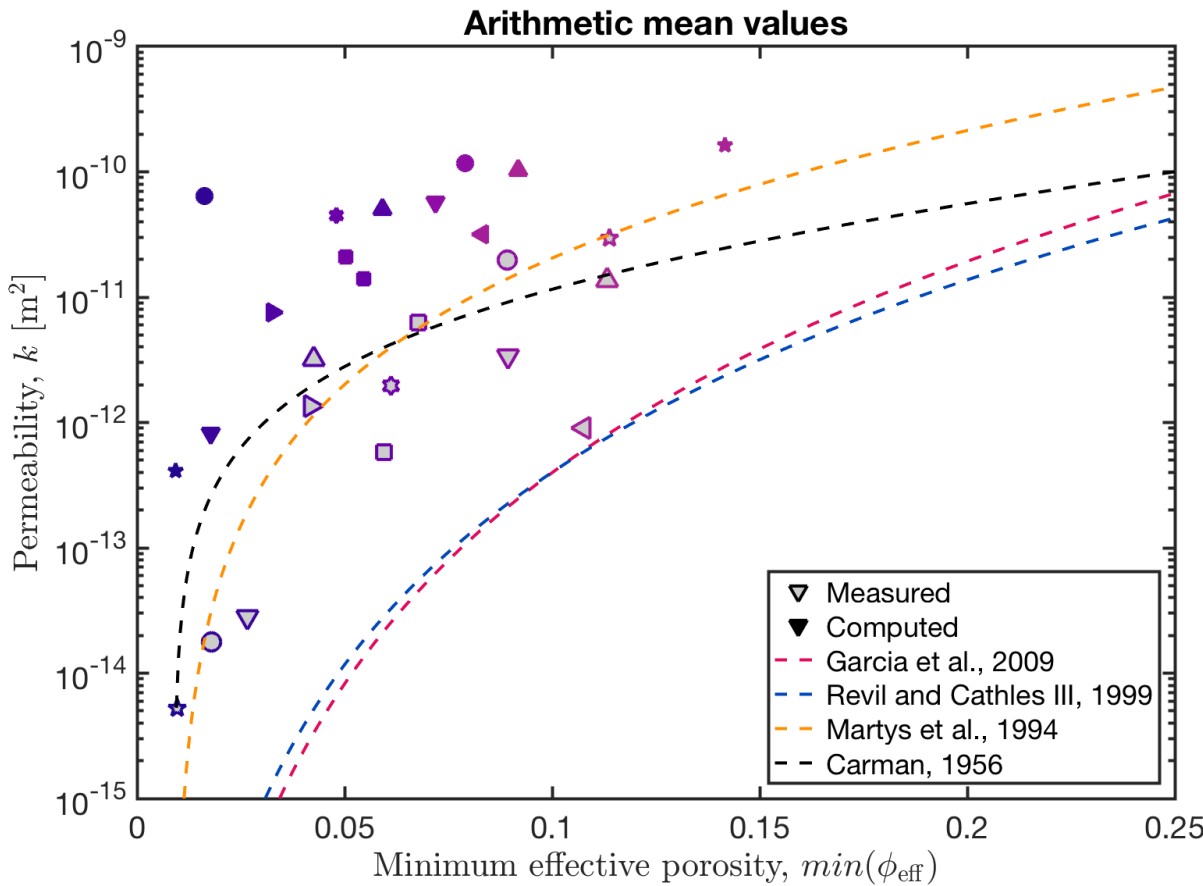

**Figure E2.** Computed and measured permeability against minimum effective porosity. Symbols of the same shape and color represent the same sample. Samples with grey face color represent measured values, whereas color only symbols stand for computed subsamples. The computed permeabilities represent the arithmetic mean values of all subsamples. To verify existing permeability parameterizations, we plotted the relations of Revil and Cathles III (1999); Garcia et al. (2009); Carman (1956) and Martys et al. (1994) against the experimental and numerical permeabilities. Note that estimated errors for the experimental permeability measurements (tab.1a) are smaller than the displayed symbols.

**Appendix F:  Resolution test**

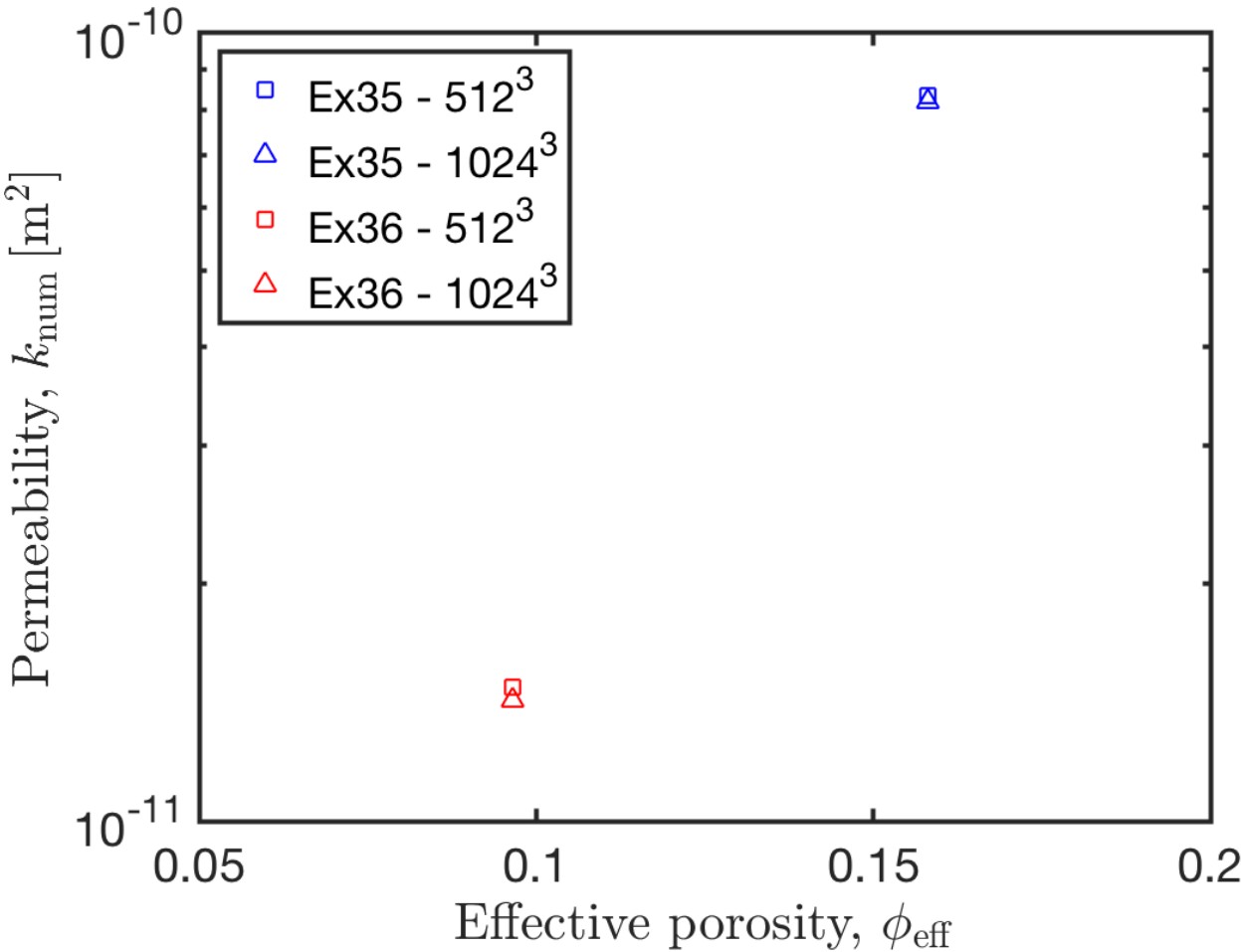

**Figure F1.** Resolution test using samples Ex35Sub04 and Ex36Sub02 (for details see also tables in the supplement). Colored squares denote standard resolution of $512^3$, whereas colored triangles are simulations with resolution of $1024^3$ voxels.

*Author contributions.* PE contributed in designing the study, sample preparation and permeability measurements. Futhermore PE did visualization, writing, methodology and running simulations. MT contributed in data interpretation, methodology, designing the study and manuscript writing. WF performed sample preparation and permeability measurements. GJG contributed in designing the study, data interpretation and manuscript writing. MN designed the study and contributed in data interpretation. SO contributed in sample preparation and measurements. TK sintered the glass bead porous media. MOK performed the resolution test. All authors contributed in writing and improving the manuscript.

*Competing interests.* The authors declare that they have no conflict of interest.

*Acknowledgements.* This work has been supported by DFG (grant no. GRK 2156/1), the JSPS Japanese-German graduate externship and BMBF (grant no. 03G0865A). M.T. was supported by the Bayerisches Geoinstitut Visitors Program. Calculations were performed on clusters btrzx2, University of Bayreuth and Mogon II, Johannes Gutenberg University Mainz. We thank Kirill Gerke and an anonymous reviewer for their constructive comments that helped to improve the manuscript considerably.

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
