# Peer review of "Combined numerical and experimental study of microstructure and permeability in porous granular media"

_Solid Earth, 2019_

## Referee Comment (RC1) · Kirill Gerke (Referee) · 9 Mar 2020

The paper is interesting and follows logically from the previous paper of the same main Author. If I understood correctly, the paper was not accepted for review by 3 potential reviewers and for this reason finally ended up with me (again). I found the idea of lab experiment and pore-scale simulations to be very relevant, we do lack such studies. But while reading this manuscript more deeply i was somewhat taken aback by Kozeny-Carman relationships the Authors use. While I find lab vs. modelling work to be very important and do support this paper to be published with SE (after some rebranding), i regret to say that I have a major point of criticism here as well. It really

puzzles me why would modern researchers utilize Kozeny-Carman relationship and why everybody at some point want to establish some kind of K-C relationship? How useful is that? We know very well already that what works for spheres does not work for real porous media samples. Moreover, the concept of hydraulic tortuosity, while still popular, provides very low information bulk measure of flow velocity field (as Authors show depending on the methodology to compute tau, the results are quite different). It may be so that computed tau values are interesting to show that they are different from previously computed, this again provides close to zero scientific value. So, while Authors proposed a "novel" Kozeny-Carman model, my question – how is it even useful, practical or simply scientifically valuable? This puts the conclusion for this work into a state of not really going anywhere. If compared against lab measurements or simulations K-C produces orders of magnitude errors, as is evident from your figures. To relate to previous results for spheres or another K-C relationship you could refer to: Martys, N. S., Torquato, S., & Bentz, D. P. (1994). Universal scaling of fluid permeability for sphere packings. Physical Review E, 50(1), 403. Garcia, X., Akanji, L. T., Blunt, M. J., Matthai, S. K., & Latham, J. P. (2009). Numerical study of the effects of particle shape and polydispersity on permeability. Physical Review E, 80(2), 021304. Now, around lines 270-275 you discuss why the results of permeability for simulations are different from these of lab measured values. While you mention that size and boundary effects could influence your results (for such small volumes i would warily estimate an error due to boundary condition to be up to 20-50%, and in this regard you could refer to Gerke, K. M., Karsanina, M. V., & Katsman, R. (2019). Calculation of tensorial flow properties on pore level: Exploring the influence of boundary conditions on the permeability of three-dimensional stochastic reconstructions. Physical Review E, 100(5), 053312), i think the main reason is different. As you can see from figure 2 you have very high porosity contrast along z-axis. Now, if you have 0.05 porosity down there – this part will dominate the porosity for the whole sample. This makes sense, as you lab values are always lower. What i would do with your (really good!) data? I would leave all this K-C and tortuosity thing, but rewrite it as not useful and your data clearly shows

that (which is, again, good). Now, you could assemble all these small pieces of 3D images you modelled with FDM solver into a 3d matrix of permeability values and upscale it (as simply as harmonic means should do the trick i suppose) to compare again the lab. This could lead to something interesting – at least you would be able to show how different model and lab values are. You could use these simple upscaling schemes as inspiration: Jang, J., Narsilio, G. A., & Santamarina, J. C. (2011). Hydraulic conductivity in spatially varying media—a pore-scale investigation. Geophysical journal international, 184(3), 1167-1179. With this little addition you paper could be completely rebranded from meaningless K-C to something really relevant to our field (kind of full core comparison between lab and modelling). Hope this helps and does not introduce too much addition work. Otherwise it is very hard for me to accept the paper as is - i think we have to automatically reject all papers dealing with K-C (just because it is wasting of time, money, pages, you name it).

Below are some additional minor comments: 1) Table 1 – is porosity measured (as computed from mass and volume?) or computed from images? How A is computed? Do all samples have the same trends in porosity as in Fig.2, if so, does porosity represent an average for the whole cylinder? 2) 2.6 – do you state that you use phi_eff for all later computations as porosity? If so, please, make it easier to guess. 3) 2.7 – how do you compute the area? By voxel counting and summarizing the interface as voxel faces? 4) Eq10-11 and Eq.12 utilize different $V\_b$ and $V\_B$ values but i guess refer to the same volume. 5) Not clear why you report Eq.14-17 if you use Eq.13 (which seems to me to be superior as it calculates hydraulic tortuosity using streamlines instead of lausy porosity-based relationships). 6) 3.2 – your model is basically the same as of Kopponen. The scatter is huge, is there any point in using such relationships? (Later I see you also substitute the points instead of this relationship, but I do not see the difference between them, is there any?) 7) Eq.23 have simply tau, not tau_H (as i guess it should be?). 8) around line 230: sorry, but i could not follow your explanation of critical exponent through, including this paragraph and also appendix D. How did you evaluated phi_c? 9) Could you, please, also describe the sample preparation procedure a

bit more, in particular how do you wrap it into resin? I could not get it completely from the current description.

---

## Referee Comment (RC2) · Anonymous Referee #2 · 24 Mar 2020

This is a neat manuscript that combines experiments and numerical models to investigate permeability in isotropic, low-porosity granular media. The authors measured permeability in sintered glass beads samples using a permeameter, and they evaluate samples' effective porosity and effective specific surface analysing CT-scan images. The results show that the values of permeability computed based on CT-scan images analysis are consistent with measured values. Finally, the authors propose a modified Kozeny-Carman equation that well predicts permeability at low porosities. Reliable predictions of permeability are of primary interest for numerical modelling of large-scale permeability, and this study contributes to its understanding, though limited to isotropic granular medium.

[Figure]

The manuscript is overall well-written. The introduction is focused, the methods are clearly described, and the results are reported in detail. However, a few points of the discussion remain unclear, in my view. Thus, I recommend the manuscript for publication in Solid Earth after the following comments will be suitably considered. These comments will hopefully help to strengthen and clarify certain aspects of the manuscript.

1) In the abstract, the authors stress the importance of characterizing fluid flow at different scales, and they state their study can be used to simulate permeability in large-scale numerical modelling. However, the up-scale of the results and the limitations of the proposed approach are never properly discussed. Therefore, it is difficult to understand how and to what extent the permeability prediction proposed in this paper is applicable to large scale modelling.

2) It is not clear how the porosity of the sintered samples is evaluated. Only through CT-scan analysis? If so, could the authors measure it experimentally (e.g., pycnometer)? This would give a measure of the effective porosity of the samples and could be compared to the computed one.

Moreover, how is the porosity reported in table 1 evaluated, both total and effective? From Figure 2, the porosity in a single sample changes quite a lot from ∼5% to ∼20% (and the reported value in table 1 is ∼13%). During permeability experiments, the low porosity zone at the bottom of the samples controls the overall permeability values resulting in a shift of the points toward higher porosity values in the permeability versus porosity plot (i.e., Figure 5). This could explain the discrepancy between computed permeability using subsamples and measured permeability of the entire sample. Could the authors add in Table 1 the minimum porosities for all the samples (or report in the supplementary material all the curves showing the height of samples versus porosities)? Could the authors plot the measured permeability versus the minimum porosity in Figure 5?

Furthermore, what is the size of subsamples in z direction? Could the author clarify it in the main text?

3) In figure 4b, the relation proposed by Koponen et al. (1996) seems to fit the data similarly to the relations proposed by the authors (Figure 4d). If I understand properly, the authors justify the choice arguing that the fits presented in Figure 4a, b and c have negative or low R2 values. However, they write that also the fit shown in Figure 4d has a low R2. The R2 values for the fits in Figure 4 are not reported in the main text. Thus, it is difficult for the reader to understand why the fit in Figure 4d is better than the fit in Figure 4c. Could the authors add this information in the main text? Could the authors clarify why they do not use Koponen et al. (1996) hydraulic tortuosity-porosity relation?

In the following, I give a few line-by-line comments:

1) line 6: The sentence "We determine flow properties like hydraulic tortuosity and permeability using both experimental measurements and numerical simulations." could be misleading. Hydraulic tortuosity is not determined by experimental measurement. Could the authors clarify it?

2) line 199: Could the authors define the hydraulic radius?

3) line 200: Is the hydraulic radius constant? Is it not affected by different porosities?

4) line 215 and line 219: Could the authors add R2 values in the text?
* * *

---

## Author Comment (AC1) · 5 May 2020

Response to comments of Kirill Gerke on the manuscript "Combined numerical and experimental study of microstructure and permeability in porous granular media" by Philipp Eichheimer et al., se-2019-199.

We thank Kirill Gerke for his great review. His constructive and useful comments that helped us to improve our manuscript.

[Figure]

Please find below a point by point response to the comments (comments of the reviewer in black and our response in blue).

Sincerely,
Philipp Eichheimer on behalf of the authors

The paper is interesting and follows logically from the previous paper of the same main Author. If I understood correctly, the paper was not accepted for review by 3 potential reviewers and for this reason finally ended up with me (again). I found the idea of lab experiment and pore-scale simulations to be very relevant, we do lack such studies. But while reading this manuscript more deeply i was somewhat taken aback by Kozeny-Carman relationships the Authors use. While I find lab vs. modelling work to be very important and do support this paper to be published with SE (after some re- branding), i regret to say that I have a major point of criticism here as well. It really puzzles me why would modern researchers utilize Kozeny-Carman relationship and why everybody at some point want to establish some kind of K-C relationship? How useful is that? We know very well already that what works for spheres does not work for real porous media samples. Moreover, the concept of hydraulic tortuosity, while still popular, provides very low information bulk measure of flow velocity field (as Authors show depending on the methodology to compute tau, the results are quite different). It may be so that computed tau values are interesting to show that they are different from previously computed, this again provides close to zero scientific value. So, while Authors proposed a "novel" Kozeny-Carman model, my question – how is it even useful, practical or simply

scientifically valuable? This puts the conclusion for this work into a state of not really going anywhere. If compared against lab measurements or simulations K-C produces orders of magnitude errors, as is evident from your figures. To relate to previous results for spheres or another K-C relationship you could refer to: Martys, N. S., Torquato, S., & Bentz, D. P. (1994). Universal scaling of fluid permeability for sphere packings. Physical Review E, 50(1), 403. Garcia, X., Akanji, L. T., Blunt, M. J., Matthai, S. K., & Latham, J. P. (2009). Numerical study of the effects of particle shape and polydispersity on permeability. Physical Review E, 80(2), 021304. Now, around lines 270-275 you discuss why the results of permeability for simulations are different from these of lab measured values. While you mention that size and boundary effects could influence your results (for such small volumes i would warily estimate an error due to boundary condition to be up to 20-50%, and in this regard you could refer to Gerke, K. M., Karsanina, M. V., & Katsman, R. (2019). Calculation of tensorial flow properties on pore level: Exploring the influence of boundary conditions on the per- meability of three-dimensional stochastic reconstructions. Physical Review E, 100(5), 053312), i think the main reason is different. As you can see from figure 2 you have very high porosity contrast along z-axis. Now, if you have 0.05 porosity down there – this part will dominate the porosity for the whole sample. This makes sense, as you lab values are always lower. What i would do with your (really good!) data? I would leave all this K-C and tortuosity thing, but rewrite it as not useful and your data clearly shows that (which is, again, good). Now, you could assemble all these small pieces of 3D images you modelled with FDM solver into a 3d matrix of permeability values and upscale it (as simply as harmonic means should do the trick i suppose) to compare again the lab. This could lead to something interesting – at least you would be able to show how different model and lab values are. You could use these simple upscaling schemes as inspiration: Jang, J., Narsilio, G. A., & Santamarina, J.

C. (2011). Hydraulic conductivity in spatially varying media - a pore-scale investigation. Geophysical journal international, 184(3), 1167-1179. With this little addition you paper could be completely rebranded from meaningless K-C to something really relevant to our field (kind of full core comparison between lab and modelling). Hope this helps and does not introduce too much addition work. Otherwise it is very hard for me to accept the paper as is - i think we have to automatically reject all papers dealing with K-C (just because it is wasting of time, money, pages, you name it).

Thank you for your detailed comment regarding the usage of the Kozeny-Carman relation. We rebranded and restructured our manu-script as suggested in your comment to not only focus on the Kozeny-Carman relation. We refrained from completely removing the Kozeny-Carman equation from the paper, as it is still frequently used in different scientific areas.

Instead, we now evaluate different published permeability parameterizations. We find that the modified Kozeny-Carman equation and the parameterization by Martys et al., 1994 provide a similarly good fit to the numerical and experimental permeability values, but also that they fail to capture second-order microstructural effects.

We also incorporated your comment on permeability upscaling and now report permeabilities not per subsample, but as the geometric mean of all subsamples.

In the methods section we now also discuss your comment regarding the minimum effective porosity controlling the permeability entire sample and modified figure 5 to account for the minimum effective porosity. We decided to keep the results on hydraulic tortuosity in the manuscript as the parameter of hydraulic tortuosity is quite important not only for the Kozeny-Carman relation but is highly interesting for several engineering disciplines.

1. Table 1 – is porosity measured (as computed from mass and volume?) or computed from images? How A is computed? Do all samples have the same trends in porosity as in Fig.2, if so, does porosity represent an average for the whole cylinder?

   1) The porosity in our study is computed from the obtained CT-images only. An experimental technique using a pycnometer, as suggested by reviewer #2, is not possible as we do not have access to such a device.
   2) The cross-sectional area A we used to determine permeability is obtained using ImageJ.
   3) Nearly all samples show a densification trend in porosity. Only samples with very low porosity do not show densification. Reported porosities are averages for the whole cylinder. We now also report the minimum effective porosity for each sample.

2. 2.6 – do you state that you use phi_eff for all later computations as porosity? If so, please, make it easier to guess.

   Thank you for this remark. We now state this issue more clearly (Page 8, line 160 ff.).

3. 2.7 – how do you compute the area? By voxel counting and summarizing the interface as voxel faces?

   We computed the area of an isosurface from the CT images using MatLab. In detail we compute an isosurface of the binary images and then the area of the resulting isosurface.

4. Eq.10-11 and Eq.12 utilize different V_b and V_B values but i guess refer to the same volume.

   We corrected the equations.

5. Not clear why you report Eq.14-17 if you use Eq.13 (which seems to me to be superior as it calculates hydraulic tortuosity using streamlines instead of lausy porosity-based relationships).

As the computation and interpretation of the hydraulic tortuosity is still under debate we wanted to give a brief overview for the reader, which relations have been proposed by other authors to which we compare our results later in figure 4. We specifically wanted to show that the porosity-based relationships do not perform well compared to our data.

6. 3.2 – your model is basically the same as of Koponen. The scatter is huge, is there any point in using such relationships? (Later I see you also substitute the points instead of this relationship, but I do not see the difference between them, is there any?)

This is a good point. We now only use the arithmetic average of the tortuosity as an input for the Kozeny-Carman equation. We also noted that using either a fit to the porosity-tortuosity relationship by Koponen or an arithmetic average only have a minor effect on predicted permeabilities. However, as hydraulic tortuosity itself (besides its potential effect on permeability) is of interest in different scientific fields, we think it is important to report our results here.

7. Eq.23 have simply tau, not tau_H (as i guess it should be?).

We corrected this mistake.

8. around line 230: sorry, but i could not follow your explanation of critical exponent through, including this paragraph and also appendix D. How did you evaluated phi_c?

We based the critical porosity threshold on porosity measurements of the samples used in our study. By systematically analyzing each sample we observed that for samples below 1% porosity we did not find any connected cluster, while samples with porosities slightly higher than 1% contained a percolating cluster. For this reason, we employed a critical porosity threshold of 0.01 instead of the published value of 0.03. As the additional description provided in Appendix D was not specifically concerned with this issue, but rather with a general explanation on why a critical porosity threshold exists, we chose to remove the appendix and the corresponding figure, as this caused too much confusion.

9. Could you, please, also describe the sample preparation procedure a bit more, in particular how do you wrap it into resin? I could not get it completely from the current description.

Thank you for your comment. In a first step we wrap the sintered glass bead sample into a high viscous resin. This can be done as this resin has a very high viscosity and can be deformed by hand. For this reason the sintered glass bead sample is literally wrapped or rolled into the resin. Just the top and bottom surface, which are needed for the experimental permeability measurements are left open. After drying, the glass bead sample with the attached highly viscous resin is embedded into a low viscous resin to create a surface, which can be sealed during the experiments. To avoid any leaks between the sample and the attached O-rings of the permeameter, both surfaces, top and bottom, are polished.

---

## Author Comment (AC2) · 5 May 2020

Response to comments of an anonymous referee on the manuscript "Combined numerical and experimental study of microstructure and permeability in porous granular media" by Philipp Eichheimer et al., se-2019-199.

We thank the anonymous referee for his review. His constructive comments helped us to improve our manuscript.

[Figure]

Please find below a point by point response to the comments (comments of the reviewer in black and our response in blue).

Sincerely,
Philipp Eichheimer on behalf of the co-authors

1. In the abstract, the authors stress the importance of characterizing fluid flow at different scales, and they state their study can be used to simulate permeability in large- scale numerical modelling. However, the up-scale of the results and the limitations of the proposed approach are never properly discussed. Therefore, it is difficult to understand how and to what extent the permeability prediction proposed in this paper is applicable to large scale modelling.

   Thank you for this comment. The proposed permeability parameterizations can be used to predict permeability on the large-scale using numerical simulations. For this reason the parameterizations are useful for isotropic low porosity media e.g. sandstones. In nature rocks mostly consists of various grain shapes and sizes, for which the proposed parameterizations are only partially valid. We now discuss this issue in the manuscript (p. 18, line 381 ff.)

2. It is not clear how the porosity of the sintered samples is evaluated. Only through CT-scan analysis? If so, could the authors measure it experimentally (e.g., pycnometer)? This would give a measure of the effective porosity of the samples and could be compared to the computed one.

Moreover, how is the porosity reported in table 1 evaluated, both total and effective? From Figure 2, the porosity in a single sample changes quite a lot from $\sim 5\%$ to $\sim 20\%$ (and the reported value in table 1 is $\sim 13\%$). During permeability experiments, the low porosity zone at the bottom of the samples controls the overall permeability values resulting in a shift of the points toward higher porosity values in the permeability versus porosity plot (i.e., Figure 5). This could explain the discrepancy between computed permeability using subsamples and measured permeability of the entire sample. Could the authors add in Table 1 the minimum porosities for all the samples (or report in the supplementary material all the curves showing the height of samples versus porosities)? Could the authors plot the measured permeability versus the minimum porosity in Figure 5?

Furthermore, what is the size of subsamples in z direction? Could the author clarify it in the main text?

Thank you for this comment. The porosity is only measured from the obtained CT-scans. Unfortunately, we do not have access to a pycnometer and therefore it is not possible to provide experimental porosity values.
The effective porosity represents all connected void clusters which contribute to the fluid flow and therefore permeability. The total porosity also takes into account inclusions and clusters which are not connected to the top and bottom of the sample.
We agree that permeability may not necessarily be affected by the total effective porosity, but rather by the minimum effective porosity in a sample (in a slice perpendicular to the flow direction). We therefore also report the minimum effective porosity of each sample and added the values in table 1 and changed figure 5 to plot permeability against the minimum effective porosity.
The height of the sample in z-direction is reported in table 1 and is around

5 mm.

3. In figure 4b, the relation proposed by Koponen et al. (1996) seems to fit the data similarly to the relations proposed by the authors (Figure 4d). If I understand properly, the authors justify the choice arguing that the fits presented in Figure 4a, b and c have negative or low R2 values. However, they write that also the fit shown in Figure 4d has a low R2. The R2 values for the fits in Figure 4 are not reported in the main text. Thus, it is difficult for the reader to understand why the fit in Figure 4d is better than the fit in Figure 4c. Could the authors add this information in the main text? Could the authors clarify why they do not use Koponen et al. (1996) hydraulic tortuosity-porosity relation?

Thank you for this comment. We added $R^2$ values to all plots for the hydraulic tortuosity.
In general, all of the proposed relations for hydraulic tortuosity do not show good agreement, in particular the ones proposing an strong increase in hydraulic tortuosity when the critical porosity is approached. The relation of Koponen et al. (1996) shows that that the value of hydraulic tortuosity does not change significantly with different porosities, thus representing a similar trend to our data. As all fits, represented by a low $R^2$ value, do not properly fit out data we used the arithemtic mean of all calculated hydraulic tortuosities for the permeability parameterization.

4. The sentence "We determine flow properties like hydraulic tortuosity and permeability using both experimental measurements and numerical simulations." could be misleading. Hydraulic tortuosity is not determined by experimental measurement. Could the authors clarify it?

This is correct, the old formulation was misleading. We modified the corresponding sentences as hydraulic tortuosity and permeability are computed numerically and the experimental permeability measurements are used to verify the obtained parameterization. (p.1, line 6 ff.)

5. Could the authors define the hydraulic radius?

We now give a definition for the hydraulic radius. (p.12, line 252)

6. Is the hydraulic radius constant? Is it not affected by different porosities?

The hydraulic radius only depends on grain size, which controls the effective pore volume between adjacent grains and is thus rather a pore-specific than a volume-specific property. As our samples consist of sintered glass bead packings with a relatively narrow grain size distribution, pore sizes throughout the sample do not vary significantly and thus also not the hydraulic radius. During sintering, some of these pores are closed, but the remaining pores do not significantly change their size. For this reason, the hydraulic radius also remains approximately constant.

7. Could the authors add R2 values in the text?

We added the corresponding $R^2$ values to the plots of hydraulic tortuosity.